# REGULARIZED LINEAR CONVOLUTIONAL NETWORKS INHERIT FREQUENCY SENSITIVITY FROM IMAGE STATISTICS

## ABSTRACT

It is widely acknowledged that trained convolutional neural networks (CNNs) have different levels of sensitivity to signals of different frequency. In particular, a number of empirical studies have documented CNNs sensitivity to low-frequency signals. In this work we show with theory and experiments that this observed sensitivity is a consequence of the frequency distribution of natural images, which is known to have most of its power concentrated in low-to-mid frequencies. Our theoretical analysis relies on representations of the layers of a CNN in frequency space, an idea that has previously been used to accelerate computations and study implicit bias of network training algorithms, but to the best of our knowledge has not been applied in the domain of model robustness.

## 1 INTRODUCTION

Since their rise to prominence in the early 1990s, convolutional neural networks (CNNs) have formed the backbone of image and video recognition, object detection, and speech to text systems (Lecun et al., 1998). The success of CNNs has largely been attributed to their "hard priors" of spatial translation invariance and local receptive fields (Goodfellow et al., 2016, §9.3). On the other hand, more recent research has revealed a number of less desirable and potentially data-dependent biases of CNNs, such as a tendency to make predictions on the basis of texture features (Geirhos et al., 2019). Moreover, it has been repeatedly observed that CNNs are sensitive to perturbations in targeted ranges of the Fourier frequency spectrum (Guo et al., 2019; Sharma et al., 2019) and further investigation has shown that these frequency ranges are dependent on training data (Yin et al., 2019; Bernhard et al., 2021; Abello et al., 2021; Maiya et al., 2022). In this work, we provide a *mathematical explanation* for these frequency space phenomena, showing with theory and experiments that neural network training causes CNNs to be most sensitive to frequencies that are prevalent in the training data distribution.

Our theoretical results rely on representing an idealized CNN in frequency space, a strategy we borrow from (Gunasekar et al., 2018). This representation is built on the classical convolution theorem,

$$\widehat{w * x} = \hat{w} \cdot \hat{x} \tag{1.1}$$

where $\hat{x}$ and $\hat{w}$ denote the Fourier transform of $x$ and $w$ respectively, and $*$ denotes a convolution. Equation 1.1 demonstrates that a Fourier transform converts convolutions into products. As such, in a "cartoon" representation of a CNN in frequency space, the convolution layers become *coordinate-wise multiplications* (a more precise description is presented in section 3). This suggests that in the presence of some form of weight decay, the weights $\hat{w}$ for high-power frequencies in the training data distribution will grow during training, while weights corresponding to low-power frequencies in the training data will be suppressed. The resulting uneven magnitude of the weights $\hat{w}$ across frequencies can thus account for the observed uneven perturbation-sensitivity of CNNs in frequency space. We formalize this argument for linear CNNs (without biases) in sections 3 and 4.

One interesting feature of the framework set up in section 4 is that the discrete Fourier transform (DFT) representation of a linear CNN is *precisely* a feedforward network with block diagonal weight matrices, where each block corresponds to a frequency index. We show in theorem 4.9 that a learning objective for such a network of depth $L$ with an $\ell_2$-norm penalty on weights is equivalent to an

objective for the associated linear model with an $\ell_p$ penalty on the singular values of each of its blocks, i.e. each frequency index — this result is new for CNNs with multiple channels and outputs. In particular, the latter penalty is highly sparsity-encouraging, suggesting as depth increases these linearly-activated CNNs have an even stronger incentive to prioritize frequencies present in the training data.

It has long been known that the frequency content of natural images is concentrated in low-to-mid frequencies, in the sense that the power in Fourier frequency $f$ is well-described by $1/|f|^\alpha$ for a coefficient $\alpha \approx 1$ (Lee et al., 2001). Hence, when specialized to training data distributions of natural images, our results explain findings that CNNs are more susceptible to low frequency perturbations in practice (Guo et al., 2019; Sharma et al., 2019).

We use our theoretical results to derive specific predictions: CNN frequency sensitivity aligns with the frequency content of training data, and deeper models, as well as models trained with substantial weight decay, exhibit frequency sensitivity more closely reflecting the statistics of the underlying images. We confirm these predictions for *nonlinear* CNNs trained on the CIFAR10 and ImageNette datasets. Figure 1 shows our experimental results for a variety of CNN models trained on CIFAR10 as well as a variant of CIFAR10 preprocessed with high pass filtering (more experimental details will be provided in section 5).

To the best of our knowledge, ours is the first work to connect the following research threads (see section 2 for further discussion):

- equivalences between linear neural networks and sparse linear models,
- classical data-dependent "shrinkage" properties of sparse linear models,
- statistical properties of natural images, and
- sensitivity of CNNs to perturbations in certain frequency ranges.

## 2 RELATED WORK

The following a brief synopsis of work most closely related to this paper; a more through survey can be found in appendix A.

**Perturbations in frequency components**: (Guo et al., 2019) found that adversarial perturbations constrained to low frequency Fourier components allowed for greater query efficiency and higher transferability between different neural networks, and (Sharma et al., 2019) demonstrated that constraining to high or or midrange frequencies did *not* produce similar effects. (Jo & Bengio, 2017), (Yin et al., 2019), (Bernhard et al., 2021), (Abello et al., 2021) and (Maiya et al., 2022) all found in one way or another that model frequency sensitivity depends on the underlying training data. Our work began as an attempt to explain this phenomenon mathematically.

**Implicit bias and representation cost of CNNs**: Our analysis of (linear) convolutional networks leverages prior work on implicit bias and representational cost of CNNs, especially (Gunasekar et al., 2018). There it was found that for a linear CNN trained on a binary linear classification task with exponential loss, with linear effective predictor $\beta$, the Fourier transformed predictor $\hat{\beta}$ converges (in direction) to a first-order stationary point of

$$\min \frac{1}{2}|\hat{\beta}|_{2/L} \text{ such that } y^n \hat{\beta}^T x^n \geq 1 \text{ for all } n. \tag{2.1}$$

Our general setup in section 3 closely follows these authors', and our theorem 4.9 partially confirms a suspicion of (Gunasekar et al., 2018, §6) that "with multiple outputs, as more layers are added, even fully connected networks exhibit a shrinking sparsity penalty on the singular values of the effective linear matrix predictor ..."

While the above result describes a form of *implicit* regularization imposed by gradient descent, we instead consider *explicit* regularization imposed by auxiliary $\ell_2$ norm penalties in objective functions, and prove equivalences of minimization problems. In this sense our analysis is technically more closely related to that of (Dai et al., 2021), which considers parametrized families of functions $f(x, w)$ and defines the *representation cost* of a function $g(x)$ appearing in the parametric family as

$$R(g) := \min\{|w|_2^2 \,|\, f(x, w) = g(x) \text{ for all } x\}. \tag{2.2}$$

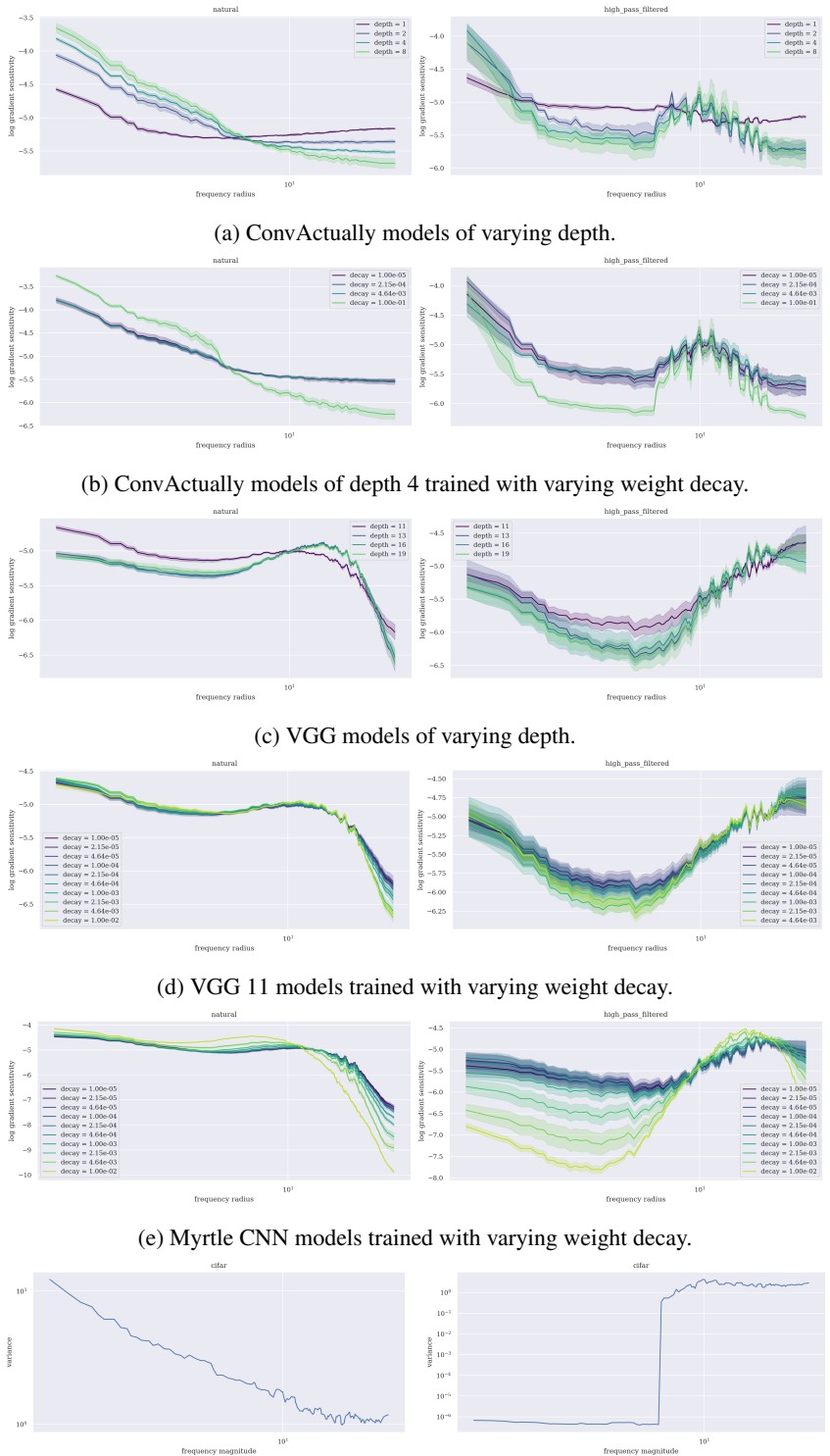

Figure 1: Radial averages $E[||\nabla_x f(x)^T \hat{e}_{cij}|| \,|\, |(i,j)| = r]$ of frequency sensitivities of CNNs trained on (hpf-)CIFAR10, post-processed by dividing each curve by its integral, taking logarithms and smoothing by averaging with 3 neighbors on either side. **Bottom row**: frequency statistics of (hpf-)CIFAR10 for comparison. See section 5 for further details.

While this approach lacks the intimate connection with the gradient descent algorithms used to train modern neural networks, it comes with some benefits: for example, results regarding representation cost are agnostic to the choice of primary loss function (e.g. squared error, cross entropy, even contrastive losses).

**Data-dependent bias**: Perhaps the work most closely related to ours is that of (Hacohen & Weinshall, 2022) on *principal component bias*, where it is shown that rates of convergence in deep (and wide) linear networks are governed by the spectrum of the input data covariance matrix.

## 3   THE DISCRETE FOURIER TRANSFORM OF A CNN

In this section we fix the notation and structures we will be working with. We define a class of idealized, linear convolutional networks and derive a useful representation of these networks in the frequency space via the discrete Fourier transform.

Consider a linear, feedforward 2D-CNN $f(x)$ of the form

$$\mathbb{R}^{C \times H \times W} \xrightarrow{w^1 *-} \mathbb{R}^{C_1 \times H \times W} \xrightarrow{w^2 *-} \mathbb{R}^{C_2 \times H \times W} \xrightarrow{w^3 *-} \cdots \xrightarrow{w^{L-1} *-} \mathbb{R}^{C_{L-1} \times H \times W} \xrightarrow{w^{L,T}-} \mathbb{R}^K \tag{3.1}$$

where $w^l * x$ denotes the convolution operation between tensors $w^l \in \mathbb{R}^{C_l \times H \times W \times C_{l-1}}$ and $x \in \mathbb{R}^{C_{l-1} \times H \times W}$, defined by

$$(w^l * x)_{cij} = \sum_{m+m'=i, n+n'=j} \Big( \sum_d w^l_{cmnd} x_{dm'n'} \Big) \tag{3.2}$$

and $w^{L,T} x$ denotes a contraction (a.k.a. Einstein summation) of the tensor $w^L \in \mathbb{R}^{K \times H \times W \times C_{L-1}}$ with the tensor $x \in \mathbb{R}^{C_{L-1} \times H \times W}$ over the last 3 indices (the $(-)^T$ denotes a transpose operation described momentarily). Explicitly,

$$(w^{L,T} x)_k = \sum_{l,m,n} w^L_{kmnl} x_{lmn}. \tag{3.3}$$

Thus, the model eq. (3.1) has weights $w_l \in \mathbb{R}^{C_l \times H \times W \times C_{l-1}}$ for $l = 1, \ldots, L-1$ and $w_L \in \mathbb{R}^{K \times H \times W \times C_{L-1}}$.

*Remarks* 3.4.   For tensors with at least 3 indices (such as $x$ and the weights $w_l$ above) we will always use the transpose notation $-^T$ to denote reversing the second and third tensor indices, which will always be the 2D spatial indices. For matrices and vectors it will be used according to standard practice. In eq. (3.3) the transpose ensures that the indices in Einstein sums move from "inside to out" as is standard practice.

Equivalently, $w^{L,T} x$ can be described as a usual matrix product $\tilde{w}_L \text{vec}(x)$ where $\text{vec}(x)$ is the vectorization (flattening) of $x$ and $\tilde{w}_L$ is obtained by flattening the last 3 tensor indices of $w_L$ (compatibly with those of $x$ as dictated by eq. (3.3)). Hence it represents a typical "flatten and then apply a linear layer" architecture component. Our reason for adopting the tensor contraction perspective is that it is more amenable to the Fourier analysis described below.

Note that in this network the number of channels is allowed to vary but the heights and widths remain fixed, and that we use full $H \times W$ convolutions throughout as opposed to the local (e.g. $3 \times 3$) convolutions often occurring in practice.

Given an array $x \in \mathbb{R}^{C_l \times H \times W}$, we may consider its discrete Fourier transform (DFT) $\hat{x}$, whose entries are computed as

$$\hat{x}_{cij} = \frac{1}{\sqrt{HW}} \sum_{m,n} x_{cmn} \exp(-\frac{2\pi \imath}{H} mi - \frac{2\pi \imath}{W} nj). \tag{3.5}$$

Similarly, for an array $w \in \mathbb{R}^{C_l \times H \times W \times C_{l-1}}$ the DFT $\hat{w}$ is defined to be

$$\hat{w}_{cijd} = \frac{1}{\sqrt{HW}} \sum_{m,n} w_{cmnd} \exp(-\frac{2\pi \imath}{H} mi - \frac{2\pi \imath}{W} nj). \tag{3.6}$$

In what follows the DFT will always be taken with respect to the two spatial dimensions and no others. The mapping $x \mapsto \hat{x}$ defines an orthogonal linear transformation of $\mathbb{R}^{C_l \times H \times W}$. In addition, it satisfies the following two properties:

**Lemma 3.7** (Cf. (Kiani et al., 2022, Lem. C.2)).

$$\widehat{w * x} = \hat{w} \cdot \hat{x} \text{ for } w \in \mathbb{R}^{C_l \times H \times W \times C_{l-1}}, x \in \mathbb{R}^{C_l \times H \times W} \text{ (convolution theorem)} \tag{3.8}$$

$$w^T x = \hat{w}^T \hat{x} \text{ for } w \in \mathbb{R}^{C_l \times H \times W \times C_{l-1}}, x \in \mathbb{R}^{C_{l-1} \times H \times W} \text{ (Parseval's theorem).} \tag{3.9}$$

Explicitly, the products on the right hand sides of eqs. (3.8) and (3.9) are defined as

$$(\hat{w} \cdot \hat{x})_{cij} = \sum_d \hat{w}_{cijd} \hat{x}_{dij} \text{ and } (\hat{w}^T \hat{x})_c = \sum_{d,i,j} \hat{w}_{cijd} \hat{x}_{dij} \text{ respectively.} \tag{3.10}$$

*Remark* 3.11. In the terminology of CNNs, this says that the DFT converts full $H \times W$ convolutions to spatially pointwise matrix multiplications, and preserves dot products.

Our first lemma is a mild generalization of (Gunasekar et al., 2018, Lem. 3); we defer all proofs to appendix D.

**Lemma 3.12.** *The CNN $f(x)$ is functionally equivalent to the network $\hat{f}(\hat{x})$ defined as*

$$\mathbb{C}^{C \times H \times W} \xrightarrow{\hat{-}} \mathbb{C}^{C \times H \times W} \xrightarrow{\hat{w}^1 \cdot -} \mathbb{C}^{C_1 \times H \times W} \cdots \xrightarrow{\hat{w}^{L-1} \cdot -} \mathbb{C}^{C_{L-1} \times H \times W} \xrightarrow{\hat{w}^{L,T} -} \mathbb{C}^K, \tag{3.13}$$

*where the first map $\hat{-}$ denotes the DFT $x \mapsto \hat{x}$.*

## 4 REGULARIZED CNN OPTIMIZATION PROBLEMS IN FREQUENCY SPACE

Consider a learning problem described as follows: given a dataset $\mathcal{D} = \{(x^n, y^n) \in \mathbb{R}^{C \times H \times W} \times \mathbb{R}^K \mid i = 1, \ldots, N\}$, and a loss function $\mathcal{L} : (\mathbb{R}^K)^N \times (\mathbb{R}^K)^N \to \mathbb{R}$, we seek weights $w$ solving the $\ell_2$-regularized minimization problem

$$\min_w \mathcal{L}((f(x^n))_{n=1}^N, (y^n)_{n=1}^N) + \lambda \sum_l |w^l|_2^2 \text{ or equivalently given lemma 3.12,}$$

$$\min_{\hat{w}} \mathcal{L}((\hat{f}(\hat{x}^n))_{n=1}^N, (y^n)_{n=1}^N) + \lambda \sum_l |\hat{w}^l|_2^2 \tag{4.1}$$

This setup allows a wide variety of loss functions. In the experiments of section 5 and appendix B we consider two important special case: first, supervised-style empirical risk minimization with respect to a sample-wise loss function $\ell : \mathbb{R}^K \times \mathbb{R}^K \to \mathbb{R}$ (in our experiments, cross entropy), where

$$\mathcal{L}((f(x^n))_{n=1}^N, (y^n)_{n=1}^N) = \frac{1}{N} \sum_{n=1}^N \ell(f(x^n), y^n) \tag{4.2}$$

Second, *contrastive* losses such as the alignment and uniformity objective of (Wang & Isola, 2020), which encourages the features $f(x^n)$ to be closely aligned with their corresponding "labels" $y^n$, and the *set* of features $\{f(x^n)\}_{n=1}^N$ to be uniformly distributed on the unit sphere $S^{K-1} \subset \mathbb{R}^K$.

According to lemma 3.12,

$$\hat{f}(\hat{x}) = \hat{w}^{L,T}(\hat{w}^{L-1} \cdots \hat{w}^1 \cdot \hat{x}). \tag{4.3}$$

In the case where the numbers of channels $C, C_1, \ldots, C_{L-1}$ are all 1 and the number of classes $K = 1$, networks of this form were studied in (Tibshirani, 2021) where they were termed "simply connected." In the case where the the number of classes $K = 1$ but the numbers of channels $C_l$ may be larger, such networks were studied in (Gunasekar et al., 2018). Of course, eq. (4.3) is just an over-parametrized linear function. We can describe it more succinctly by introducing a new tensor $\hat{v} \in \mathbb{R}^{K \times H \times W \times C}$ such that $\hat{f}(\hat{x}) = \hat{v}^T \hat{x}$. With a little manipulation of eq. (4.3), we can obtain a formula for $\hat{v}$ in terms of the $\hat{w}_l$ for $l = 1, \ldots, L$.

**Lemma 4.4.**

$$\hat{v} = \hat{w}^{L,T} \cdot \hat{w}^{L-1} \cdots \hat{w}^1 \tag{4.5}$$

The following theorem shows that the regularization term of eq. (4.1), which penalizes the $\ell_2$-norms of the factors $\hat{w}^l$, is equivalent to a penalty using more sparsity-encouraging norms of $\hat{v}$. To state it we need a definition.

**Definition 4.6** (Schatten $p$-norms). Let $A = (a_{ij}) \in M(n \times n, \mathbb{C})$ be a square matrix and let $A = UDV^T$ be a singular value decomposition of $A$, where $U, V \in U(n)$ and $D = \text{diag}(\lambda_i)$ is a non-negative diagonal matrix with diagonal entries $\lambda_1, \ldots, \lambda_n \geq 0$. For any $p > 0$ the **Schatten $p$-norm of** $A$ is

$$\|A\|_p^S = (\sum_n |\lambda_i|^p)^{\frac{1}{p}}. \tag{4.7}$$

*Remark* 4.8. In the case $p = 2$, the 2-norm of definition 4.6 agrees with the usual Euclidean 2-norm $(\sum_{i,j} |a_{ij}|^2)^{\frac{1}{2}}$, since left (resp. right) multiplication by a unitary matrix $U^T$ (resp. $V$) preserves the Euclidean 2-norms of the columns (resp. rows) of $A$.

**Theorem 4.9.** *The optimization problem eq. (4.1) is equivalent to an optimization problem for $\hat{v}$ of the form*

$$\min_{\hat{v}} \mathcal{L}((\hat{v}^T \hat{x}^n)_{n=1}^N, (y^n)_{n=1}^N) + \lambda L \sum_{i,j} (\|\hat{v}_{ij}\|_{\frac{2}{L}}^S)^{\frac{2}{L}}, \tag{4.10}$$

*where[1] $\hat{v}_{ij}$ denotes the $K \times C$ matrix obtained by fixing the spatial indices of $\hat{v} = (\hat{v}_{cijd})$, and the $\min$ is taken over the space of tensors $\hat{v}$ such that each matrix $\hat{v}_{ij}$ has rank at most $\min\{C, C_1, \ldots, C_{L-1}, K\}$.*

The essential ingredient of our proof is a generalized non-commutative Hölder inequality.

**Lemma 4.11.** *If $B \in M(m \times n, \mathbb{C})$ is a matrix with complex entries, $A_1, \ldots, A_k$ is a composable sequence of complex matrices such that $A_L \cdots A_1 = B$ and $\sum_i \frac{1}{p_i} = \frac{1}{r}$ where $p_1, \ldots, p_L, r > 0$ are positive real numbers,*

$$\|B\|_r^S \leq \prod_i \|A_i\|_{p_i}^S \tag{4.12}$$

Such inequalities are not new: in the case $r = 1$, lemma 4.11 follows from (Dixmier, 1953, Thm. 6), and in the case $L = 2$ it is an exercise in (Bhatia, 1996). However, we suspect (and our proof of theorem 4.9 suggests) that lemma 4.11 underpins many existing results on implicit bias and representation costs of (linear) neural networks, such as those of (Gunasekar et al., 2018; Yun et al., 2021; Dai et al., 2021; Lawrence et al., 2022; Jagadeesan et al., 2022).

In the case where the numbers of channels $C, C_1, \ldots, C_{L-1}$ are all 1 and and the number of outputs $K = 1$, *and* where the loss $\ell$ is squared error, the problem eq. (4.10) reduces to

$$\min_{\hat{v}} \frac{1}{N} \sum_n |y^n - \sum_{i,j} \hat{v}_{ij} \hat{x}_{ij}|_2^2 + \lambda L \sum_{i,j} |\hat{v}_{ij}|^{\frac{2}{L}}. \tag{4.13}$$

The sum in the regularization term of eq. (4.13) is $|\hat{v}|_p^p$ where $p = \frac{2}{L}$ — in particular when $L = 1$, eq. (4.13) is a ridge regression problem (Hastie et al., 2001; Tibshirani & Wasserman) and when $L = 2$ (the one hidden layer case) eq. (4.13) is a LASSO problem. We can analyze these two tractable cases to obtain qualitative predictions which will be tested empirically in section 5. Since the qualitative predictions from both cases are similar, we devote the following section to ridge, and defer LASSO to appendix C.

### 4.1 $L = 1$: RIDGE REGRESSION

In this case, eq. (4.13) is the usual ridge regression objective; the closed-form solution is

$$(\lambda + \frac{1}{N} \hat{X}^T \hat{X}) \hat{v} = \frac{1}{N} \hat{X}^T Y \tag{4.14}$$

where $\hat{X}$ is a $N \times C \times H \times W$ batch tensor with "rows" the $\hat{x}^n$ and the entries of $Y$ are the $y^n$ (see e.g. (Hastie et al., 2001)). When $\lambda = 0$ this reduces to the usual (unpenalized) least squares solution $\hat{v}_{LS} := (\hat{X}^T \hat{X})^{-1} \hat{X}^T Y$, and substituting $\hat{X}^T Y = \hat{X}^T \hat{X} \hat{v}_{LS}$ in eq. (4.14) we obtain

$$(\lambda + \frac{1}{N} \hat{X}^T \hat{X}) \hat{v} = \frac{1}{N} \hat{X}^T \hat{X} \hat{v}_{LS} \tag{4.15}$$

---

[1]by an abuse of notation for which we beg your forgiveness.

where strictly speaking $\hat{X}^T \hat{X}$ is a tensor product of $\hat{X}$ with itself in which we contract over the batch index of length $N$, hence it is of shape $W \times H \times C \times C \times H \times W$.

The frequency properties of images enter into the structure of the symmetric tensor $\frac{1}{N}\hat{X}^T\hat{X}$, which (if the dataset $\mathcal{D}$ is centered, i.e. preprocessed by subtracting the mean $\frac{1}{N}X^T\mathbf{1}_N$) serves as a generalized *covariance matrix* for the frequency space representation of $\mathcal{D}$. To ease notation, let $\Sigma = \frac{1}{N}\hat{X}^T\hat{X}$, and *suppose* that

$$\Sigma_{whcc'h'w'} \approx \begin{cases} \tau_{cij} & \text{if } (c, i, j) = (c', i', j') \\ 0 \text{ otherwise.} \end{cases} \tag{4.16}$$

In other words, proper covariances between distinct frequency components are negligible and we retain only the *variances*, i.e. the diagonal entries of the covariance matrix. In fig. 4 we demonstrate that this assumption is not unrealistic in the case where $X$ is a dataset of natural images.

With the assumption of eq. (4.16), eq. (4.15) reduces to

$$\hat{v}_{cij} = \frac{\tau_{cij}}{\lambda + \tau_{cij}}\hat{v}_{\text{LS},cij} = \frac{1}{1 + \frac{\lambda}{\tau_{cij}}}\hat{v}_{\text{LS},cij} \text{ for all } cij. \tag{4.17}$$

Equation (4.17) is an instance of the classic fact that ridge regression shrinks coefficients more in directions of low input variance. In words, $\tau_{cij}$ is the variance of training images in Fourier component $cij$, and eq. (4.17) says $|\hat{v}_{cij}|$ shrinks more when the variance $\tau_{cij}$ is low; in the limiting case $\tau_{cij} \to 0$ the coefficient $\hat{v}_{cij} \to 0$ as well.

Returning to the subject of frequency sensitivity, observe that $\hat{v}_{cij}$ is the directional derivative of $f$ with respect to the $cij$-th Fourier component.

**Proposition 4.18** (Data-dependent frequency sensitivity, $L = 1$). *With the notations and assumptions introduced above, the magnitude of the directional derivative of $f$ with respect to the $cij$-th Fourier component scales with $\lambda$ and $\tau$ according to $\frac{1}{1+\frac{\lambda}{\tau_{cij}}}$.*

Empirically it has been found that for natural distributions of images the variances $\tau_{cij}$ follow a power law of the form $\tau_{cij} \approx \frac{\gamma}{|i|^\alpha+|j|^\beta}$ (Lee et al., 2001; Baradad et al., 2021).[2] Under this model, eq. (4.17) becomes

$$\hat{v}_{cij} = \frac{1}{1 + \frac{\lambda}{\gamma}(|i|^\alpha + |j|^\beta)}\hat{v}_{\text{LS},cij} \text{ for all } cij, \tag{4.19}$$

that is, sensitivity is monotonically decreasing with respect to both frequency magnitude and the regularization coefficient $\lambda$. This is consistent with findings that CNNs trained on natural images are vulnerable to low frequency perturbations (Guo et al., 2019; Sharma et al., 2019).

## 5 EXPERIMENTS

When $L > 2$ (that is, when there are more than 1 convolutional layers), the $p$ "norm" of eq. (4.10) is non-convex (hence the quotes), and in the limit as $L \to \infty$ the norms appearing in eq. (4.10) converge to the Schatten 0-"norm," which is simply the number of non-0 singular values of a matrix.[3] Moreover, we see that the regularization coefficient of eq. (4.10) is effectively multiplied by $L$.

Even in the case where $K = 1$ so that $\hat{v}$ is a vector, it is known that solving eq. (4.10) for $L > 1$ is NP-hard (Chen et al., 2017), so we have no hope of finding closed form solutions as in section 4.1. However, we can use the analysis in section 4 to derive three testable hypotheses:

I. CNN frequency sensitivity depends on the frequency content of training data (proposition 4.18).
II. The fact that the regularization term of eq. (4.10) becomes more sparsity-encouraging as $L$ increases suggests that the data-dependent frequency sensitivity observed in section 4.1 and appendix C becomes *even more pronounced* as the number of convolutional layers increases.

---

[2] In particular, under this approximation $\tau_{cij}$ is independent of $c$.

[3] In the special case where $K = 1$ the regularization term in eq. (4.10) is the penalty of the *subset selection* problem in the field of sparse linear models.

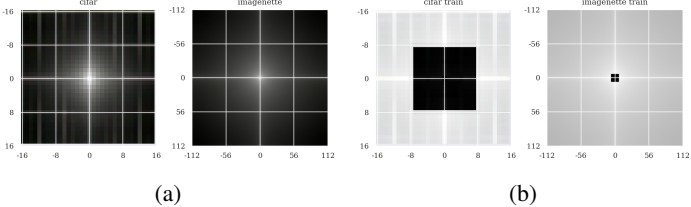

(a)                                                                                     (b)

Figure 2: **(a)** Standard deviations of image datasets along Fourier basis vectors: in the notation above, these are the $\sqrt{\Sigma_{jiccij}}$ (viewed in log scale as RGB images). The origin corresponds to the lowest frequency basis vectors (i.e. constant images). **(b)** Same as (a), but with the addition of high pass filtering removing frequency indices $(i, j)$ with $|i|, |j| \leq 8$ (we also experiment with a higher frequency cutoff on ImageNette in appendix B).

III. Moreover, the functional forms of eqs. (4.17) and (C.7) suggest that the data-dependent frequency sensitivity will increase monotonically with the weight decay parameter $\lambda$.

We empirically validate these hypotheses with experiments using CNNs trained on multiple datasets. The datasets used in these experiments are CIFAR10 (Krizhevsky, 2009), ImageNette (the 10 easiest classes from ImageNet (Deng et al., 2009)) (FastAI, 2019), and synthetic images generated using the wavelet marginal model (WMM) of (Baradad et al., 2021). The later dataset is of interest since the generative model is *explicitly* designed to capture the frequency statistics of natural images, and allows for varying the exponents $\alpha$ and $\beta$ in the power law $\tau_{cij} \approx \frac{\gamma}{|i|^\alpha + |j|^\beta}$ described above. Figure 2a display the variances CIFAR10, ImageNette and their high pass filtered variants in frequency space; for those of the WMM datasets see fig. 3. In addition, we experiment with *high pass filtered* versions of the CIFAR10 and ImageNette datasets, which we refer to as hpf-CIFAR10 and hpf-ImageNette respectively; the frequency space statistics of these are shown in fig. 2b.[4] For implementation details we refer to appendix B. We can further summarize the data displayed in figs. 2a and 2b by averaging over circles with varying radii $r$, i.e. computing expectations $E[\sqrt{\Sigma_{jiccij}} \mid |(i, j)| = r]$, to obtain the frequency *magnitude* statistics curves shown in fig. 1 (see appendix B.2 for implementation details). From this point forward we focus on such curves with respect to frequency magnitude.

The CNNs used in these experiments are **ConvActually** (a CNN that closely approximates eq. (3.1), the differences being the addition of biases and ReLU non-linearities), **Myrtle CNN** (a small feed-forward CNN known to achieve high performance on CIFAR10 obtained from (Page)), **VVG** (a family of CIFAR10 VGGs (Simonyan & Zisserman, 2015) obtained from (Fu, 2017) and ImageNet VGGs from (Marcel & Rodriguez, 2010)), **ResNet** (a family of CIFAR10 ResNets (He et al., 2016) obtained from (Team, 2021)), and **AlexNet** (a small AlexNet (Krizhevsky et al., 2012) adapted to contrastive training obtained from (Baradad et al., 2021)) For more detailed descriptions of datasets and model architectures we refer to appendices B.1 and B.5.

We measure frequency sensitivity of a CNN $f$ in terms of the magnitudes of the directional derivatives $\nabla_x f(x)^T \hat{e}_{cij}$, where $\hat{e}_{cij}$ is the $cij$-th Fourier basis vector. These magnitudes are averaged over all the input images $x$ in the relevant validation set, and as in the case of image statistics we can average them over circles of varying radii, i.e. compute expectations $E[|\nabla_x f(x)^T \hat{e}_{cij}| \mid |(i, j)| = r]$ (see appendix B.3 for implementation details). By "data-dependent frequency sensitivity" we mean the extent to which the frequency sensitivity of a CNN $f$ reflects the statistics of the images it was trained on. Figures 1, 2a, 3 and 5 show that the variance of DFTed CIFAR10, ImageNette and WMM images is heavily concentrated at low frequencies, in agreement with the power law form $\tau_{cij} \approx \frac{\gamma}{|i|^\alpha + |j|^\beta}$ described in section 4.[5] Hence in the absence of any modifications to the underlying images, we expect that training on this data will emphasize sensitivity of $f$ to perturbations along the lowest frequency Fourier basis vectors, with the effect increasing along with model depth and the weight decay parameter $\lambda$. On the other hand, the variance of hpf-CIFAR10 and hpf-ImageNette is concentrated in mid-range frequencies, and so here we expect training will emphasize sensitivity of $f$ to perturbations along mid range Fourier basis vectors (again with more pronounced effect as depth/$\lambda$ increase).

---

[4] This was inspired by the experiments of (Jo & Bengio, 2017) and (Yin et al., 2019).

[5] In the case of WMM this is by design.

## 5.1 FREQUENCY SENSITIVITY AND DEPTH

Figure 1a shows sensitivity of ConvActually models of varying depth to perturbations of varying DFT frequency magnitudes. These curves illustrate that as depth increases, frequency sensitivity $E[||\nabla_x f(x)^T \hat{e}_{cij}|| \,|\, |(i,j)| = r]$ more and more closely matches the frequency magnitude statistics $E[\sqrt{\Sigma_{jiccij}} \,|\, |(i,j)| = r]$ of the training data set, *both* in the case of natural and high pass filtered images — hence, these empirical results corroborate hypotheses I. and II.. Figure 1c shows results of a similar experiment with VGG models of varying depth; models trained on natural images have sensitivity generally decreasing with frequency magnitude, whereas those trained on high pass filtered data have "U" shaped sensitivity curves with minima near the filter cutoff, corroborating hypothesis I.. Here it is not clear whether models trained on natural images follow that pattern predicted by II. (the deepest models are less sensitive to both low and high frequencies), however when trained on high pass filtered images deeper models do seem to most closely follow the frequency statistics of the dataset. Figure 5 includes a similar experiment with VGG models of varying depth trained on (hpf-)ImageNette.

## 5.2 FREQUENCY SENSITIVITY AND WEIGHT DECAY

Figure 1b shows radial frequency sensitivity curves for ConvActually models of depth 4 trained on (hpf-)CIFAR10 with varying weight decay coefficient $\lambda$. We see that as $\lambda$ increases, model frequency sensitivity more and more closely reflects the statistics of the training data images, corroborating hypotheses I. and III.. Figure 1b shows results for a similar experiment with Myrtle CNNs, with a similar conclusion. Figure 5 shows results for VGG models trained with varying weight decay on (hpf-)ImageNette.

## 5.3 IMPACT OF THE LEARNING OBJECTIVE

So far, our analysis and experiments have only shown that the regularization term in eq. (4.1) *encourages* CNN gradients with respect to spatial Fourier basis vectors to reflect the frequency statistics of the training data. It is of course possible that the first term of eq. (4.1) defining the learning objective overwhelms the regularization term resulting in different model frequency sensitivity. In fig. 8 we show this occurs in CNNs trained on WMM synthetic data with an alignment and uniformity contrastive loss; see appendix B.4 for a possible explanation in the framework of sections 3 and 4.

# 6 LIMITATIONS AND OPEN QUESTIONS

In order to obtain an optimization problem with some level of analytical tractibility, we made many simplifying assumptions in sections 3 and 4, most notably omitting nonlinearities from our idealized CNNs. While the experimental results of section 5 illustrate that multiple predictions derived from sections 3 and 4 hold true for CNNs more closely resembling those used in practice trained with supervised learning, fig. 7 shows that hypotheses I-III can fail in the presence of residual connections — see appendix B.4 for further discussion. As previously mentioned, fig. 8 shows that a contrastive alignment and uniformity learning objective results in far different CNN representations.

Perhaps more significantly, it must be emphasized that model sensitivity as measured by gradients represents a very small corner of a broader picture of model robustness (or lack therof). For example, it does not encompass model behaviour on corruptions (see e.g. (Hendrycks & Dietterich, 2019)) or shifted distributions (see e.g. (Recht et al., 2019)).

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

## A    RELATED WORK IN GREATER DETAIL

**CNN sensitivity to Fourier frequency components**: (Jo & Bengio, 2017) computed transfer accuracy of image classifiers trained on data preprocessed with various Fourier filtering schemes (e.g. train on low pass filtered images, test on unfiltered images or vice versa). They found significant generalization gaps, suggesting that models trained on images with different frequency content learned different patterns.

(Guo et al., 2019) proposed algorithms for generating adversarial perturbations constrained to low frequency Fourier components, finding that they allowed for greater query efficiency and higher transferability between different neural networks. (Sharma et al., 2019) demonstrated empirically that constraining to high or or midrange frequencies did *not* produce similar effects, suggesting convolutional networks trained on natural images exhibit a particular sensitivity to low frequency perturbations.

(Yin et al., 2019) showed different types of corruptions of natural images (e.g. blur, noise, fog) have different effects when viewed in frequency space, and models trained with different augmentation strategies (e.g. adversarial training, gaussian noise augmentation) exhibit different sensitivities to perturbations along Fourier frequency components. (Diffenderfer et al., 2021) investigates the relationship between frequency sensitivity and natural corruption robustness for models compressed with various weight pruning techniques, and introduces ensembling algorithms where the frequency statistics of a test image are compared to those of various image augmentation methods, and models trained on the augmentations most spectrally similar to the test image are used in inference. (Sun et al., 2022) designs an augmentation procedure that explicitly introduces variation in both the amplitude and phase of the DFT of input images, finding it improves certified robustness and robustness and common corruptions. (Bernhard et al., 2021) investigated the extent to which constraining models to use only the lowest (or highest) Fourier frequency components of input data provided perturbation robustness, also finding significant variability across datasets. (Abello et al., 2021) tested the extent to which CNNs relied on various frequency bands by measuring model error on inputs where certain frequencies were removed, again finding a striking amount of variability across datasets. (Maiya et al., 2022) analyzed the sensitivity of networks to perturbations in various frequencies, finding significant variation across a variety of datasets and model architectures. All of these works suggest that model frequency sensitivity depends heavily on the underlying training data. — our work began as an attempt to explain this phenomenon mathematically.

**Implicit bias and representation cost of CNNs**: Our analysis of (linear) convolutional networks leverages prior work on implicit bias and representational cost of CNNs, especially (Gunasekar et al., 2018). There it was found that for a linear one-dimensional convolutional network where inputs and all hidden layers have one channel (in the notation of section 3, $C = C_1 = \cdots = C_{L-1} = K = 1$) trained on a binary linear classification task with exponential loss, with linear effective predictor $\beta$, the Fourier transformed predictor $\hat{\beta}$ converges in direction to a first-order stationary point of an optimization problem of the form

$$\min \frac{1}{2}|\hat{\beta}|_{2/L} \text{ such that } y^n \hat{\beta}^T x^n \geq 1 \text{ for all } n. \tag{A.1}$$

A generalization to arbitrary group-equivariant CNNs (of which the usual CNNs are a special case) appears in (Lawrence et al., 2022, Thm. 1) — while we suspect that some of our results generalize

to more general equivariant networks we leave that to future work. For generalizations in different directions see (Lyu & Li, 2020; Yun et al., 2021), and for additional follow up work see (Jagadeesan et al., 2022). Our general setup in section 3 closely follows these authors', and our theorem 4.9 partially confirms a suspicion of (Gunasekar et al., 2018, §6) that "with multiple outputs, as more layers are added, even fully connected networks exhibit a shrinking sparsity penalty on the singular values of the effective linear matrix predictor ..."

While the aforementioned works study the *implicit* regularization imposed by gradient descent, we instead consider *explicit* regularization imposed by auxiliary $\ell_2$ norm penalties in objective functions, and prove equivalences of minimization problems. In this sense our analysis is perhaps more closely related to that of (Dai et al., 2021), which considers parametrized families of functions $f(x, w)$ and defines the *representation cost* of a function $g(x)$ appearing in the parametric family as

$$R(g) := \min\{|w|_2^2 \mid f(x, w) = g(x) \text{ for all } x\}. \tag{A.2}$$

While this approach lacks the intimate connection with the gradient descent algorithms used to train modern neural networks, it comes with some benefits: for example, results regarding representation cost are agnostic to the choice of per-sample loss function (in particular they apply to both squared error and cross entropy loss). In the case where the number of channels $C = C_1 = \cdots = C_{L-1} = 1$ (but the number of outputs may be $> 1$), theorem 4.9 can be deduced from (Dai et al., 2021, Thm. 3).

Lastly, while in this paper we focus on *spatial frequency* properties of image data, there is a large and growing body of work on the impact of frequency properties of training data more broadly interpreted. (Rahaman et al., 2019) gave a formula for the *continuous* Fourier transform of a ReLU network $f : \mathbb{R}^n \to \mathbb{R}$, and showed in a range of experiments that ReLU networks learn low frequency modes of input data first. (Xiao, 2022) proves theoretical results on low frequency components being learned first for networks $f : \prod_i S^{n_i} \to \mathbb{R}$ on products of spheres, where the role of frequency is played by spherical harmonic indices (see also (Xiao & Pennington, 2022) for some related results).

Perhaps the work most closely related to ours is that of (Hacohen & Weinshall, 2022) on *principal component bias*, where it is shown that rates of convergence in deep (and wide) linear networks are governed by the spectrum of the input data covariance matrix.[6] (Hacohen & Weinshall, 2022) also includes experiments connecting PC bias with spectral bias (learning low frequency modes first, as in the preceding paragraph) and a phenomenon known as *learning order consistency*. However, it is worth noting that in their work there is no explicit theoretical analysis of CNNs and no consideration of the statistics of natural images in Fourier frequency space.

**Other applications of Fourier transformed CNNs**: (Zhu et al., 2021; Pratt et al., 2017; Mathieu et al., 2014; Vasilache et al., 2015) all, in one way or another, leverage frequency space representations of convolutions to accellerate computations, e.g. neural network training. Since this is not our main focus, we omit a more detailed synopsis.

# B EXPERIMENTAL DETAILS

## B.1 DATASETS

Figure 3 illustrates the frequency content of the image datasets used in this paper.

For CIFAR10 we use canonical train/test splits (imported using (Marcel & Rodriguez, 2010)).

As described at (FastAI, 2019),

> Imagenette is a subset of 10 easily classified classes from Imagenet (tench, English springer, cassette player, chain saw, church, French horn, garbage truck, gas pump, golf ball, parachute).

We use the "full size" version of the dataset with standard (224-by-224) ImageNet preprocessing.

We implement high pass filtering by passing each image $x$ through the following preprocessing steps (before any other preprocessing other than loading a JPEG image file as a tensor): (i) take the DFT $\hat{x}$, (ii) multiply with a *mask* $m$ where $m_{cij} = 0$ if $|i|, |j| \leq$ some fixed threshold, and 1

---

[6]They also prove a result for shallow ReLU networks.

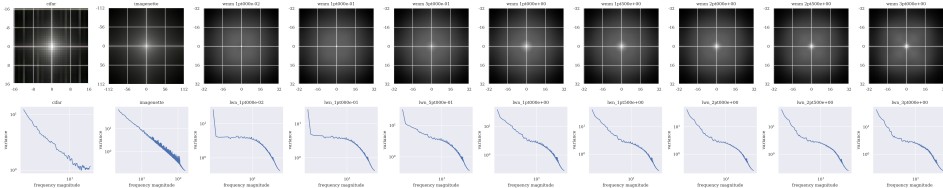

Figure 3: **Top row:** Standard deviations of image datasets (including those generated by the wavelet marginal model) along Fourier basis vectors: in the notation above, these are the $\sqrt{\Sigma_{jiccij}}$ (viewed in log scale as RGB images). The origin corresponds to the lowest frequency basis vectors (i.e. constant images). **Bottom row:** Same data as the top row, further processed by taking radial averages $E[\sqrt{\Sigma_{jiccij}} \,|\, |(i,j)| = r]$, and smoothing by averaging with three nearest neighbors on either side; shown in log-log-scale.

otherwise,[7] (iii) applying the inverse DFT. For *both* CIFAR10 and ImageNette, our threshold is 8 pixels. This means that while 25% of frequency indices are filtered out for CIFAR10, only about 0.13% are filtered for ImageNette. The motivation for this approach was to remove a similar amount of absolute frequency content in both cases; we also experimented with a threshold of 112 in the case of ImageNette, resulting in removal of 25% of frequency indices in both cases, and results of these experiments are shown in fig. 6.

We use the wavelet marginal model dataset generated using the implementation in (Baradad et al., 2021) — details of this model are described in (Baradad et al., 2021, §3.3). The generated images are of resolution 128-by-128; they are preprocessed with downsampling to 96-by-96 and cropping to 64-by-64. For further preprocessing details we refer to (Baradad et al., 2021, §4).

All pipelines described in this paper implement the standard preprocessing step of subtracting the mean RGB value of the training dataset, and dividing by the standard deviation of the training set RGB values.[8] Note that this does *not* flatten the variances of the image distributions in frequency space displayed in figs. 1 and 5 — in fact, it (provably) only impacts the variance of the 0th Fourier component, corresponding to constant images. Ensuring that the variance in each Fourier component is (approximately) 1 would require the far less standard preprocessing step to each image $x$ consisting of (i) apply the DFT to get $\hat{x}$, (ii) subtract $\hat{\mu}$, where $\hat{\mu}$ is the mean of the DFTed images in the training dataset (*not* just the mean of their RGB values),[9] (iii) divide by the standard deviation $\hat{\sigma}$ of the DFTed images in the training dataset, and finally (iv) apply the inverse DFT. It should be emphasized that $\hat{\mu}$ and $\hat{\sigma}$ have the same shape as $x$, e.g. $(3, 32, 32)$ for the CIFAR10 dataset. The only work we are aware of that implements such preprocessing on images is (Hacohen & Weinshall, 2022, see §4), though of course there may be others.

### B.2 COVARIANCE MATRICES OF DFTED IMAGE DATASETS

In this section we provide further details on our computations of (co)variances and standard deviations of DFTed image datasets.

To compute the covariances $\Sigma_{jicc'i'j'}$ for fig. 4, we begin with a dataset of natural images, say $X$. We subtract its mean RGB pixel value (a vector in $\mathbb{R}^3$) and divide by the standard deviation of RGB pixel values (also a vector in $\mathbb{R}^3$) as is standard. We next apply the DFT to every image in $X$, to obtain a DFTed dataset $\hat{X}$. We then compute the mean $\hat{\mu}$ of the *images* in $\hat{X}$, not the RGB pixel values — this is a $3 \times H \times W$ tensor, where $H, W$ are the heights/widths of the images in $X$ (e.g., 32 for CIFAR10). This mean is then subtracted from $\hat{X}$ to obtain a centered dataset. Next, we sample batches of size $B$, say $\hat{x}_1, \ldots, \hat{x}_B$, from $\hat{X}$ (these are tensors of shape $B \times C \times H \times W$). For each batch, we subtract the mean, contract over the batch index and divide by $B$ to obtain an estimate for

---

[7]Here we assume that the lowest frequency component is at the origin, so the low frequency components are zeroed out and the high frequency ones pass through.

[8]For example, in the case of ImageNet these are the canonical `[0.485, 0.456, 0.406]` and `[0.229, 0.224, 0.225]` respectively.

[9]Equivalently $\hat{\mu}$ is the DFT of the mean training image.

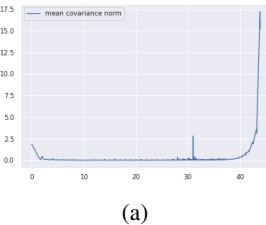 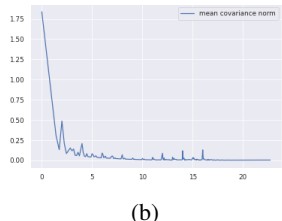

(a)                                             (b)

Figure 4: Testing the assumption of eq. (4.16) on the CIFAR10 dataset. **(a)** Average covariance entry norm $E[|\Sigma_{jicc'i'j'}|\,|\,|(c,i,j) - (c,i',j')| = r]$ plotted with respect to $r = |(i,j) - (i',j')|$. **(b)** Same as (a), except the $i,j$ part of the distance $r$ is computed *modularly*, i.e. on the discrete torus $\mathbb{Z}/H \times \mathbb{Z}/W$ instead of the discrete square $\{0,\dots,H-1\} \times \{0,\dots,W-1\}$. For more details see appendix B.

$\Sigma$, say $\tilde{\Sigma}$ like so:

$$\tilde{\Sigma} = \frac{1}{B}\sum_{b=1}^{B}(\hat{x}_b - \hat{\mu})^T \otimes (\hat{x}_b - \hat{\mu}); \text{ explicitly } \tilde{\Sigma}_{jicc'i'j'} = \frac{1}{B}\sum_{b=1}^{B}(\hat{x}_{b,jic} - \hat{\mu}_{jic})(\hat{x}_{b,c'i'j'} - \hat{\mu}_{c'i'j'})$$

(B.1)

Finally, we average over an entire dataset's worth of batches to get our final estimate of $\Sigma_{jicc'i'j'}$.

To estimate the expectations $E[|\Sigma_{jicc'i'j'}|\,|\,|(c,i,j) - (c,i',j')| = r]$, we average absolute values $|\Sigma_{jicc'i'j'}|$ over all indices $jicc'i'j'$ satisfying $|(c,i,j) - (c',i',j')| = r$. The number of such entries varies significantly with $r$, which is why we have not explicitly written down the average. We compute "modular" distances, i.e. distances on the discrete torus, using the formulae

$$d(i,j') = \min\{|(h - h')\mathrm{mod}H|, |H - ((h - h')\mathrm{mod}H)|\}$$
$$d(w,w') = \min\{|(w - w')\mathrm{mod}W|, |W - ((w - w')\mathrm{mod}W)|\}$$

(B.2)

and finally $d((c,h,w),(c',h',w')) = \sqrt{(c - c')^2 + d(h,h')^2 + d(w,w')^2}$.

In fig. 2a, we begin with a dataset of natural images, say $X$. We subtract its mean RGB pixel value (a vector in $\mathbb{R}^3$) and divide by the standard deviation of RGB pixel values (also a vector in $\mathbb{R}^3$) as is standard. We next apply the DFT to every image in $X$, to obtain a DFTed dataset $\hat{X}$. We then compute the standard deviation of the *images* in $\hat{X}$, not the RGB pixel values — this is a $3 \times H \times W$ tensor, where $H, W$ are the heights/widths of the images in $X$ (e.g., 32 for CIFAR10).

### B.3 GRADIENT SENSITIVITY IMAGES

To compute the sensitvity curves in figs. 1, 5, 7 and 8, we subsample 5,000 images from the underlying validation dataset in the case of CIFAR10 and WMM use the entire validation dataset in the case of ImageNette. For each such image $x$, we compute the DFT $\hat{x}$, and then backpropagate gradients through the composition $\hat{x} \to \hat{\hat{x}} = x \to f(x)$. The result is a $C \times H \times W \times K$ Jacobian matrix expressing the derivative of $f$ with respect to Fourier basis vectors. We take $\ell_2$ norms over the class index (corresponding to $K$) to obtain a $C \times H \times W$ gradient norm image, with $c, h, w$ component $|\nabla_x f(x)^T \hat{e}_{cij}|$. Finally, we average these gradient norms over the (subsampled in the case of CIFAR10 and WMM) dataset.

Next, we average radially much as we did in fig. 4. Given an expected gradient norm image $E[|\nabla_x f(x)^T \hat{e}_{cij}|]$ obtained as above, we further average over all indices $cij$ such that $\sqrt{i^2 + j^2} = r$. Again, the number of such indices is highly variable.

To provide error estimates, for each set of trained model weights and each image dataset, we compute radial frequency sensitivity curves as in the paragraph above, apply post-processing consisting of:

- Dividing the curve by its integral (to obtain a probability distribution)[10]

---

[10]this results in a comparison between models that is *scale invariant*, that is, we want to compare the shapes of frequency sensitivity curves, not their overall magnitudes

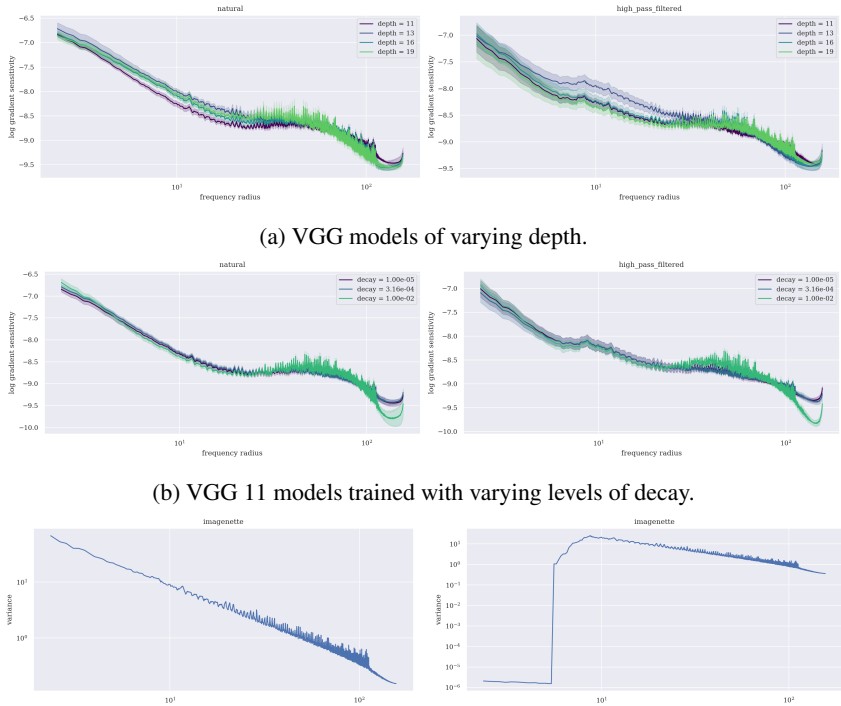

(a) VGG models of varying depth.

(b) VGG 11 models trained with varying levels of decay.

Figure 5: Radial averages $E[||\nabla_x f(x)^T \hat{e}_{cij}|| \, | \, |(i,j)| = r]$ of frequency sensitivities of VGG models trained on ImageNette and its high pass filtered variant, post processed as discussed in appendix B.3. **Bottom row**: frequency statistics of ImageNette and its high pass filtered variant for comparison.

- Taking the logarithm of the resulting probability distribution. This step is motivated by the empirical observation that the variance of Fourier transformed images follows a power law with respect to frequency magnitude.

- Smoothing by averaging with 3 nearest neighbors on each side. This step is motivated by the aforementioned highly variable number of DFT frequency indices corresponding to a given radius, which means that some radius values correspond to an average over far fewer samples which as a result has high variance. A possible alternative would have been to *bin* radii, thus averaging over DFT components with frequency magnitudes in a small interval. However, the automatic binning strategies we tried yielded bins that were too large, giving an undesirably low resolution view of frequency sensitivity curves.

We then repeat the entire pipeline above for 5 sets of model weights trained from independent random initializations, to obtain 5 curves, and display the standard deviation of their $y$-values. For more details on our training procedures, see appendix B.5.

### B.4    MORE EXPERIMENTAL RESULTS

Figure 5 shows radial frequency sensitivity curves from experiments training VGGs with variable depth and weight decay on ImageNette, with and without high pass filtering. Here we do see significant differences between models trained on natural vs. high pass filtered images, including in the later case (small) local peaks in near the filter cutoff, however the effect of filtering is not nearly as noticeable as in the CIFAR10 experiments of fig. 1.

Moreover, effects of depth and decay are not as dramatic in these experiments, although we do see the curves corresponding to high depth/decay for VGGs trained on natural images dropping off most severely at high frequencies, and in the case of high pass filtered images deep VGGs are the least sensitive in the low frequency range (where the training images have no variance).

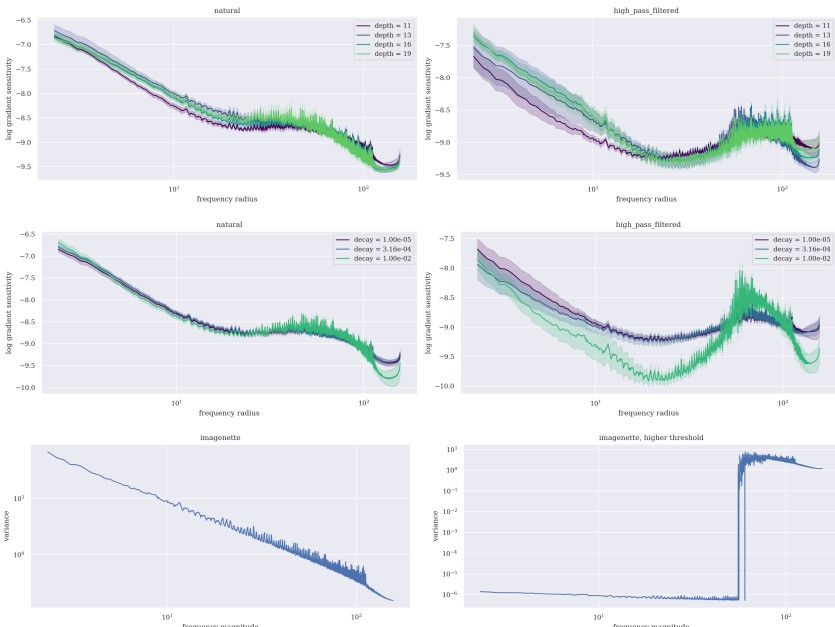

Figure 6: VGGs trained on hpf-ImageNette with a larger cutoff (removing 25% of frequency indices).

Interestingly, we see a range of frequency radii, roughly $r \in [10, 100]$, where the ImageNette frequency statistics exhibit significant noise around the overall power law pattern, and it does appear that all VGG models concentrate frequency sensitivity in this range, and more so with greater depth/decay. It would be interesting to understand what sorts of natural image features contribute to the observed noise in the $r \in [10, 100]$ range, and if they are for some reason useful to the ImageNette classification task.

Figure 6 shows a similar experiment with VGGs trained on (hpf-)ImageNette, but with a high-pass threshold of 112, so that 25% of frequency indices are discarded. Here the differences between CNNs trained on natural and high pass filtered images are (not surprisingly) more dramatic, and the effect of decay on VGG 11s trained on hpf-ImageNette is especially pronounced. It is (at best) unclear whether the results for VGGs of variable depth trained on hpf-ImageNette support hypothesis II. (depth).

Figure 7 shows radial frequency sensitivity curves from experiments training ResNets with variable depth and weight decay on CIFAR10, with and without high pass filtering. The frequency sensitivity curves are clearly different when trained on natural images (where they drop off at the highest frequencies) versus high pass filtered images (where they have a "U"-shape similar to those of the VGG models trained on high pass filtered CIFAR10 in fig. 1). These observations seems somewhat consistent with hypothesis I. (data-dependent frequency sensitivity). However, the trends with depth and decay do *not* conform with hypotheses II., III. (depth, decay): it is difficult to see trends in the depth experiment, and in the decay experiment the frequency sensitivity curves seem to become more *increasing* as decay increases, rather than adhering to the frequency content of the dataset. It is not immediately clear to the authors what causes this behavior — further investigation would be an interesting direction for future work.

Lastly, fig. 8 shows frequency sensitivity curves for AlexNets trained with alignment and uniformity loss (Wang & Isola, 2020) on synthetic data generated by the wavelet marginal model (WMM) of (Baradad et al., 2021) with varying $\alpha$ parameter; to be specific, for simplicity we set $\alpha = \beta$ in the power law $\tau_{cij} \approx \frac{\gamma}{|i|^\alpha + |j|^\beta}$. These results initially surprised us: for all $\alpha$ the variance of the synthetic data is concentrated in low frequencies (fig. 3), with the level of concentration decreasing as $\alpha$ increases (i.e. as $\alpha$ increases, the frequency curve of the synthetic data spreads out). *However*, for all AlexNets trained on WMM data the frequency sensitivity curves *increase* with frequency magnitude, with slope roughly *increasing* with respect to $\alpha$!

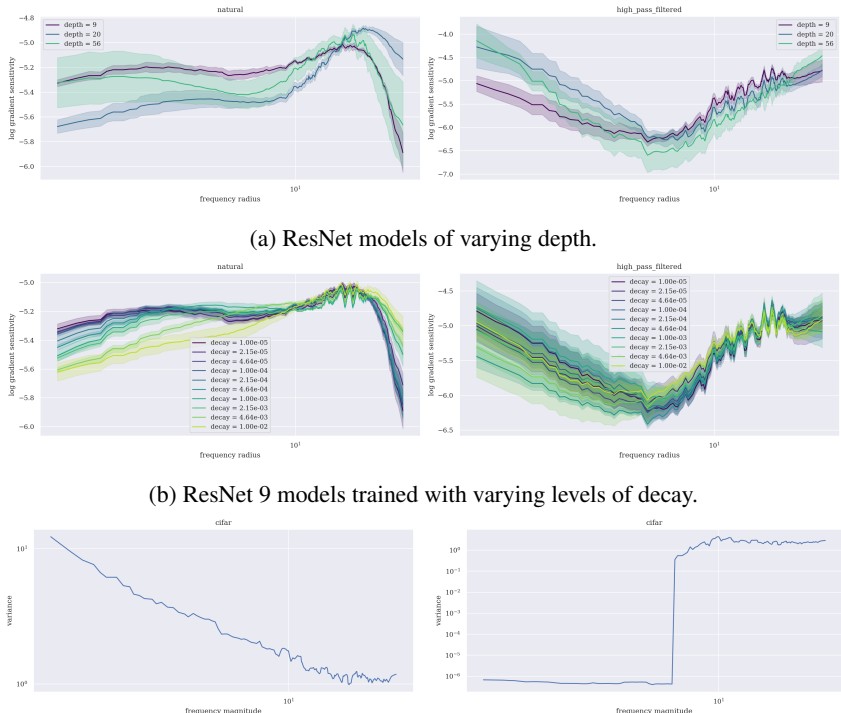

Figure 7: Radial averages $E[|\nabla_x f(x)^T \hat{e}_{cij}| \,|\, |(i,j)| = r]$ of frequency sensitivities of ResNet models of varying depth (top row) and decay (middle row) trained on CIFAR10 and its high pass filtered variant. Post processing is as discussed in appendix B.3.

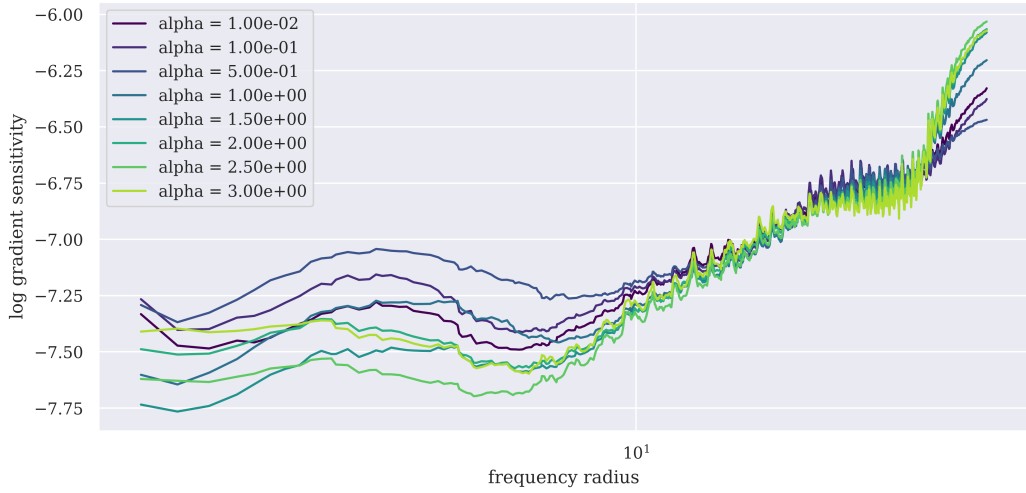

Figure 8: Radial averages $E[|\nabla_x f(x)^T \hat{e}_{cij}| \,|\, |(i,j)| = r]$ of frequency sensitivities of unsupervised AlexNet models trained with alignment and uniformity loss on data generated by WMM models with varying $\alpha$ parameter. Post-processing is as discussed in appendix B.3.

This result, which shows that CNNs trained with contrastive learning objectives can respond quite differently to the statistics of their training data, admits a simple explanation in terms of sections 3 and 4.

Indeed, the uniformity part of the objective encourages the set of vectors

$$\{\hat{f}(\hat{x^n}) = \hat{v}^T \hat{x}^n \,|\, n = 1, \ldots, N\} \tag{B.3}$$

to be uniformly distributed on the unit sphere $S^{K-1} \subseteq \mathbb{R}^K$. Suppose now for the sake of simplicity that that the vectors $\hat{x}^n = (\hat{x}_{cij}^n)$ are normally distributed and their covariance matrix $\Sigma$ is diagonal, with diagonal entries of the form

$$\tau_{cij} \approx \frac{\gamma}{|i|^\alpha + |j|^\alpha}. \tag{B.4}$$

Then the feature vectors $\hat{f}(\hat{x^n}) = \hat{v}^T \hat{x}^n$ are also normally distributed, with covariance matrix

$$\hat{v}^T \Sigma \hat{v} = \sum_{ij} \frac{\gamma}{|i|^\alpha + |j|^\alpha} \hat{v}_{ij} \hat{v}_{ij}^T. \tag{B.5}$$

Here as above $\hat{v}_{ij}$ is a $K \times C$ matrix, so each $\hat{v}_{ij}\hat{v}_{ij}^T$ is indeed a $K \times K$ positive semi-definite matrix, and in cases of interest where $C \ll K$ it will have very low rank. Considering eq. (B.5), we see for example that if $C \ll K$ and the $\hat{v}_{ij}$ are of roughly constant magnitude, then the covariance of the features $\hat{f}(\hat{x^n})$ will be dominated by the terms

$$\frac{\gamma}{|i|^\alpha + |j|^\alpha} \hat{v}_{ij} \hat{v}_{ij}^T \text{ for small } ij \tag{B.6}$$

making it impossible for the $\hat{f}(\hat{x^n})$ to be uniformly distributed on the unit sphere. It seems that perhaps the *only* way for such uniform distribution to occur is for the magnitudes of the $\hat{v}_{ij}$ to increase with the frequency magnitude $|(i, j)|$, in such a way as to offset the denominator $|i|^\alpha + |j|^\alpha$. We do not have a proof of this fact (it seems such a proof would have to involve analysis of the functional form of the contrastive loss in (Wang & Isola, 2020), which we have not carried out), but it does offer one potential explanation of fig. 8. At the risk of being overzealous, the above discussion would suggest that

$$|\hat{v}_{ij}| \propto |i|^\alpha + |j|^\alpha, \tag{B.7}$$

i.e. the norm of the gradient of $f$ with respect to the $ij$-th DFT basis vector, follows a power law in $\alpha$, and indeed in fig. 8, where these gradient norms are plotted with respect to frequency magnitude on a log-log scale, we see slope roughly increasing with $\alpha$.

### B.5 MODEL ARCHITECTURES, TRAINING PARAMETERS AND VALIDATION ACCURACIES

Recall that our CNNs are

**ConvActually** A CNN that closely approximates eq. (3.1), the differences being the addition of biases and ReLU non-linearities. This is accomplished by applying convolutions with $32 \times 32$ kernels (i.e., kernels of the same size as the input images) with circular padding.

**Myrtle CNN** A small feed-forward CNN known to achieve high performance on CIFAR10 obtained from (Page). This CNN has small kernels, ReLU non-linearities and max-pooling, as well as exponentially varying channel dimension.

**VVG** A family of CIFAR10 VGGs obtained from (Fu, 2017) and ImageNet VGGs from (Marcel & Rodriguez, 2010).

**ResNet** A family of CIFAR10 ResNets (He et al., 2016) obtained from (Team, 2021).

**AlexNet** A small AlexNet adapted to contrastive training obtained from (Baradad et al., 2021).

Tables 1 and 2 describe training hyperparameters. In all experiments we optimize using stochastic gradient descent (SGD) with momentum 0.9, using a "reduce on plateau" learning rate schedule where the learning rate is multiplied by 0.1 after "patience" epochs without a 1% improvement in validation accuracy, where "patience" is some fixed integer (i.e., a hyperparameter); we use patience = 20 throughout. This schedule proceeds until either a minimum learning rate (in all of our experiments, $10^{-6}$) or a maximum number of epochs is hit, at which point training stops. We use the PyTorch library (Paszke et al., 2019) on a cluster environment with Nvidia GPUs.

All models are trained on CIFAR10 except the ImageNette VGGs and AlexNets. The ImageNette VGGs are optimized as above, with the exception that we use distributed training on 8 GPUs (the batch size of 256 corresponds to a batch of size 32 on each GPU). The AlexNets are trained using an unsupervised alignment and uniformity objective as described in (Baradad et al., 2021, §4, §A); we use the official implementation of (Baradad et al., 2021). For CIFAR10 and ImageNette, we use the same hyperparameters when training on natural and high pass filtered images.

For each model architecture and choice of hyperparameters,[11] i.e. instance appearing in tables 1 and 2, and each training dataset, we train 5 models from different random weight initializations. Efficiently training this many CNNs was facilitated by the excellent FFCV library (Leclerc et al., 2022).

| arch. family | depth | batch size | initial lr | decay |
|---|---|---|---|---|
| Conv Actually | 4 | 1024 | $10^{-4}$ | `10**linspace(-5, -1, 4)` |
| Myrtle CNN | N/A | 512 | $10^{-2}$ | `10**linspace(-5, -2, 10)` |
| CIFAR VGG | 11 | 256 | $10^{-2}$ | `10**linspace(-5, -2, 10)` |
| ImageNette VGG | 11 | 256 | $1.250 \times 10^{-3}$ | `10**linspace(-5, -2, 3)` |
| ResNet | 9 | 512 | $10^{-1}$ | `10**linspace(-5, -2, 10)` |

Table 1: Hyperparameters of CNNs of variable decay used in the experiments of section 5.2. In all cases, the maximum number of epochs was 500.

| arch. family | depth | batch size | initial lr | max epochs |
|---|---|---|---|---|
| Conv Actually | 1,2,4,8 | 1024 | $10^{-4}$ | 500 |
| CIFAR VGG | 11, 13, 16, 19 | 256 | $10^{-2}$ | 500 |
| ImageNette VGG | 11, 13, 16 , 19 | 256 | $1.250 \times 10^{-3}$ | 500 |
| ResNet | 9, 20, 56 | 512 | $10^{-1}$ | 200 |

Table 2: Hyperparameters of CNNs of variable depth used in the experiments of section 5.1. In all cases weight decay was $10^{-5}$.

## C    $L = 2$: LASSO PARAMETER SHRINKING AND SELECTION

When $C_l = 1$ for all $l$ and $L = 1$, we may simplify eqs. (4.10) and (4.13) to

$$\min_{\hat{v}} \frac{1}{N}|Y - \hat{X}\hat{v}|_2^2 + 2\lambda|\hat{v}|_1 \tag{C.1}$$

where $\hat{X}$ and $Y$ are as in section 4.1. The optimality criterion for eq. (4.10) becomes (see e.g. (Tibshirani & Wasserman))

$$\frac{1}{N}(\hat{X}^T\hat{X}\hat{v} - \hat{X}^TY) + \lambda\nabla|\hat{v}|_1 = 0, \tag{C.2}$$

where $\nabla|\hat{v}|_1$ is the *sub-gradient* of the $\ell_1$-norm:

$$(\nabla|\hat{v}|_1)_i \begin{cases} = \text{sign}(\hat{v}_i) & \text{if } \hat{v}_i \neq 0 \\ \in [-1, 1] & \text{if } \hat{v}_i = 0. \end{cases} \tag{C.3}$$

---

[11]*Except for* the AlexNets trained on WMM data.

When $\lambda = 0$ this again reduces to the unpenalized least squares solution $\hat{v}_{\text{LS}} := (\hat{X}^T \hat{X})^{-1} \hat{X}^T Y$, and substituting this in eq. (C.2) we obtain

$$\frac{1}{N}(\hat{X}^T \hat{X} \hat{v} - \hat{X}^T \hat{X} \hat{v}_{\text{LS}}) + \lambda \nabla |\hat{v}|_1 = 0. \tag{C.4}$$

If we again make the assumption that $\Sigma = \frac{1}{N} X^T X$ is "diagonal" as in eq. (4.16), eq. (C.2) simplifies to

$$\tau_{cij}(\hat{v}_{cij} - \hat{v}_{\text{LS},cij}) + \lambda \frac{d|\hat{v}_{cij}|}{d\hat{v}_{cij}} = 0 \text{ for all } cij, \tag{C.5}$$

where strictly speaking $\frac{d|\hat{v}_{cij}|}{d\hat{v}_{cij}}$ is a subgradient as in eq. (C.3). From this we conclude

$$\hat{v}_{cij} \begin{cases} = \hat{v}_{\text{LS},cij} - \frac{\lambda}{\tau_{cij}} \operatorname{sign}(\hat{v}_{cij}), & \text{if } \hat{v}_{cij} \neq 0 \\ \in [\hat{v}_{\text{LS},cij} - \frac{\lambda}{\tau_{cij}}, \hat{v}_{\text{LS},cij} + \frac{\lambda}{\tau_{cij}}] & \text{if } \hat{v}_{cij} = 0 \end{cases} \text{ for all } cij \tag{C.6}$$

(the second case is equivalent to: if the LASSO solution $\hat{v}_{cij} = 0$, it must be that the least squares solution satisfies $|\hat{v}_{\text{LS},cij}| \leq \frac{\lambda}{\tau_{cij}}$). In the case where $\operatorname{sign}(\hat{v}_{cij}) = \operatorname{sign}(\hat{v}_{\text{LS},cij})$, we obtain a particularly nice conclusion:

$$|\hat{v}_{cij}| = |\hat{v}_{\text{LS},cij}| - \frac{\lambda}{\tau_{cij}} \text{ for all } cij \tag{C.7}$$

**Proposition C.8** (Data-dependent frequency sensitivity, $L = 2$). *With the notations and assumptions introduced above, the magnitude of the directional derivative of $f$ with respect to the $cij$-th Fourier component is linear in $\frac{\lambda}{\tau_{cij}}$ with slope $-1$.*

If we again plug in the empirically determined power law observed in natural imagery, $\tau_{cij} \approx \frac{\gamma}{|i|^\alpha + |j|^\beta}$, eq. (C.7) becomes

$$|\hat{v}_{cij}| = |\hat{v}_{\text{LS},cij}| - \frac{\lambda}{\gamma}(|i|^\alpha + |j|^\beta) \text{ for all } cij. \tag{C.9}$$

Here, the sensitivity decreases monotonically with respect to both frequency magnitude and the regularization coefficient $\lambda$. Note that compared to eq. (4.19) from section 4, eq. (C.9) implies a greater shrinking effect when $|\hat{v}_{cij}| < 1$, and less shrinking when $|\hat{v}_{cij}| > 1$.[12] We conjecture that shrinkage due to $|\hat{v}_{cij}| < 1$ is the dominant effect, due to the initial distribution of weights $w$ in modern neural networks, which are often sampled from uniform or normal distributions with variance $\ll 1$ (e.g., according to He initialization; He et al., 2016).

# D PROOFS

## D.1 PROOF OF LEMMAS 3.7, 3.12 AND 4.4

*Proof of lemma 3.7.* For a direct proof of the convolution part, see (Kiani et al., 2022, Lem. C.2).

We reduce to the "single channel" cases of these formulae ($C_l = C_{l-1} = 1$), which we take to be well known. For the purposes of legibility, in this proof we denote the DFT (eqs. (3.5) and (3.6)) by $\mathcal{F}$.

For the convolution part we must apply the DFT to eq. (3.2). Let $w_{c,\dots,d}$ denote the single channel tensor obtained by fixing the input/output channels to indices $c, d$ and similarly let $x_{d,\dots}$ be the single channel tensor fixing the input channel to index $d$. Then by interchanging the order of sums

$$\begin{aligned} (w * x)_{cij} &= \sum_{m+m'=i, n+n'=j} \left( \sum_d w_{cmnd} x_{dm'n'} \right) \\ &= \sum_d \left( \sum_{m+m'=i, n+n'=j} w_{cmnd} x_{dm'n'} \right) \\ &= \sum_d \left( (w_{c,\dots,d} * x_{d,\dots})_{ij} \right) \end{aligned} \tag{D.1}$$

---

[12]This remark applies generally to LASSO vs. ridge regression, and is perhaps most easily explained by comparing the derivatives of $|x|$ and $x^2$.

In other words, $(w * x)_{c,...} = \sum_d w_{c,...,d} * x_{d,...}$ Now linearity of the DFT gives

$$\mathcal{F}((w * x)_{c,...}) = \sum_d \mathcal{F}(w_{c,...,d} * x_{d,...}) = \sum_d \mathcal{F}(w_{c,...,d}) \cdot \mathcal{F}(x_{d,...}) \tag{D.2}$$

where on the right hand side we've applied the standard single channel convolution theorem. But this means

$$\mathcal{F}(w * x)_{cij} = \sum_d \mathcal{F}(w)_{cijd}\mathcal{F}(x)_{dij} \tag{D.3}$$

as desired.

Our proof of the Parseval identity is similar: we start with eq. (3.3), that is, $(w^T x)_k = \sum_{l,m,n} w_{kmnl}x_{lmn}$. With notation as above, we can write

$$(w^T x)_k = \sum_{l,m,n} w_{kmnl}x_{lmn} = \sum_l w_{k,...,l}^T x_{l,...} \tag{D.4}$$

where each term on the right hand side is an ordinary inner product of single channel signals. Taking the Parseval identity for these as well known, we get

$$(w^T x)_k = \sum_l \mathcal{F}(w)_{k,...,l}^T \mathcal{F}(x)_{l,...} = \sum_{l,m,n} \mathcal{F}(w)_{kmnl}^T \mathcal{F}(x)_{lmn} \tag{D.5}$$

as desired. □

*Proof of lemmas 3.12 and 4.4.* We proceed by induction on $L$. In the base case where $L = 0$, lemma 3.12 is equivalent to Parseval's theorem eq. (3.9). So, suppose $L > 0$. We may decompose $f$ as a convolution followed by a linear CNN of the form eq. (3.1), say $g$, with one fewer layers. Explicitly, $g$ has weights $w_2, \ldots, w_L$ and

$$f(x) = g(w_1 * x). \tag{D.6}$$

By inductive hypothesis, we may assume that

$$g(w_1 * x) = \hat{g}(\widehat{w_1 * x}) = \hat{w}^{L,T}\Big(\big(\prod_{l=1}^{L-1} \hat{w}_{L-l,...ij}\big)\widehat{w_1 * x}_{:,ij}\Big) \tag{D.7}$$

Applying the convolution theorem eq. (3.8) to obtain

$$\widehat{w_1 * x}_{:ij} = \hat{w}_{1,...,ij}\hat{x}_{:,ij} \tag{D.8}$$

completes the proof. □

*Proof of lemma 4.4.* For each weight $\hat{w}^l$ let $\hat{w}_{ij}^l$ denote the $C_l \times C_{l-1}$ matrix obtained by fixing the spatial indices of $\hat{w}^l$ to $ij$. Unpacking definitions,

$$\big(\hat{w}^{L-1} \cdot \hat{w}^{L-1} \cdots \hat{w}^1 \cdot \hat{x}\big)_{ij} = \hat{w}_{ij}^{L-1} \cdot \hat{w}_{ij}^{L-1} \cdots \hat{w}_{ij}^1 \cdot \hat{x}_{ij} \text{ and so}$$
$$\hat{w}^{L,T}\big(\hat{w}^{L-1} \cdot \hat{w}^{L-1} \cdots \hat{w}^1 \cdot \hat{x}\big)_c = \sum_{i,j} \sum_d (\hat{w}_{ij}^L)_{cd}\big(\hat{w}_{ij}^{L-1} \cdot \hat{w}_{ij}^{L-1} \cdots \hat{w}_{ij}^1 \cdot \hat{x}_{ij}\big)_d \tag{D.9}$$

We can recognize the inside sum as performing a matrix product of $\hat{w}_{ij}^L$ with $\hat{w}_{ij}^{L-1} \cdot \hat{w}_{ij}^{L-1} \cdots \hat{w}_{ij}^1 \cdot \hat{x}_{ij}$. Since matrix multiplication is associative, we can just as well multiply the $\hat{w}_{ij}^l$ first and *then* act on the vector $\hat{x}_{ij}$. Thus

$$\sum_{i,j} \sum_d (\hat{w}_{ij}^L)_{cd}\big(\hat{w}_{ij}^{L-1} \cdot \hat{w}_{ij}^{L-1} \cdots \hat{w}_{ij}^1 \cdot \hat{x}_{ij}\big)_d = \sum_{i,j} \sum_d \big(\hat{w}_{ij}^L \cdot \hat{w}_{ij}^{L-1} \cdot \hat{w}_{ij}^{L-1} \cdots \hat{w}_{ij}^1\big)_{cd} \cdot (\hat{x}_{ij})_d \tag{D.10}$$

Now we can recognize the right hand side as $(\hat{w}^L \cdot \hat{w}^{L-1} \cdots \hat{w}^1)^T \hat{x}$, as claimed. □

## D.2 PROOFS OF THEOREM 4.9 AND LEMMA 4.11

Recall we aim to prove: the optimization problem

$$\min_{\hat{w}} \mathcal{L}(((\hat{w}^L \cdot \hat{w}^{L-1} \cdot \hat{w}^{L-1} \cdots \hat{w}^1)^T \hat{x}^n)_{n=1}^N, (y^n)_{n=1}^N) + \lambda \sum_{l,i,j} |\hat{w}_{ij}^l|_2^2 \tag{D.11}$$

is equivalent to the following optimization problem for the product $\hat{v}_{\cdots ij} = \prod_{l=0}^L \hat{w}_{L-l,\ldots ij}$:

$$\min_{\hat{v}} \mathcal{L}((\hat{v}^T \hat{x}^n)_{n=1}^N, (y^n)_{n=1}^N) + \lambda L \sum_{i,j} (\|\hat{v}_{ij}\|_{\frac{2}{L}}^S)^{\frac{2}{L}}. \tag{D.12}$$

where the minima runs over the space of tensors $\hat{v}$ such that each matrix $\hat{v}_{ij}$ has rank at most $\min\{C, C_1, \ldots, C_{L-1}, K\}$. We proceed by a series of reductions; as a first step we observe

$$\min_{\hat{w}} \mathcal{L}(((\hat{w}^L \cdot \hat{w}^{L-1} \cdot \hat{w}^{L-1} \cdots \hat{w}^1)^T \hat{x}^n)_{n=1}^N, (y^n)_{n=1}^N) + \lambda \sum_{l,i,j} |\hat{w}_{ij}^l|_2^2$$

$$= \min_{\hat{v}} \min_{\hat{w}^L \cdot \hat{w}^{L-1} \cdot \hat{w}^{L-1} \cdots \hat{w}^1 = \hat{v}} \mathcal{L}((\hat{v}^T \hat{x}^n)_{n=1}^N, (y^n)_{n=1}^N) + \lambda \sum_{l,i,j} |\hat{w}_{ij}^l|_2^2, \tag{D.13}$$

and hence theorem 4.9 will follow if we can prove

$$\sum_{i,j} (\|\hat{v}_{ij}\|_{\frac{2}{L}}^S)^{\frac{2}{L}} = \min \frac{1}{L} \sum_{l,i,j} |\hat{w}_{ij}^l|_2^2, \tag{D.14}$$

where the min on the right hand side runs over all $\hat{w}$ such that $\hat{w}^L \cdot \hat{w}^{L-1} \cdot \hat{w}^{L-1} \cdots \hat{w}^1 = \hat{v}$. We can make life slightly simpler by noticing eq. (D.14) decomposes over the $i, j$ index, and will follow from

$$(\|\hat{v}_{ij}\|_{\frac{2}{L}}^S)^{\frac{2}{L}} = \min \frac{1}{L} \sum_{l} |\hat{w}_{ij}^l|_2^2 \text{ for all } i, j, \tag{D.15}$$

which we will show in lemma D.18 — note that at this point we are arriving at a statement about norms of matrix products and can dispense with the baggage of $i, j$ indices and ˆs. To formally state that lemma, we introduce a convenient definition.

**Definition D.16.** A sequence of matrices $A_1 \in M(m_1 \times n_1, \mathbb{C}), \ldots, A_L \in M(m_L \times n_L, \mathbb{C})$ is **composable** if and only if

$$m_l = n_{l+1} \text{ for } l = 1, \ldots, L-1 \tag{D.17}$$

In other words, $A_1, \ldots, A_L$ is composable if and only if the product $A_L \cdots A_1$ makes sense.

**Lemma D.18.** *If $B \in M(m \times n, \mathbb{C})$ is a matrix with complex entries and $L \in \mathbb{N}$ is a non-negative integer, then*

$$(\|B\|_{\frac{2}{L}}^S)^{\frac{2}{L}} = \min(\prod_l |A_l|_2^2)^{\frac{1}{L}} = \min \frac{1}{L} \sum_l |A_l|_2^2, \tag{D.19}$$

*where both minima are taken over all composable sequences of complex matrices $A_1, \ldots, A_L$ such that $A_L \cdots A_1 = B$.*

We first deal with the elementary aspects of lemma D.18: it will suffice to show that whenever $A_L \cdots A_1 = B$,

$$(\|B\|_{\frac{2}{L}}^S)^{\frac{2}{L}} \leq (\prod_l |A_l|_2^2)^{\frac{1}{L}} \leq \frac{1}{L} \sum_l |A_l|_2^2 \text{ and that} \tag{D.20}$$

$$(\|B\|_{\frac{2}{L}}^S)^{\frac{2}{L}} = \frac{1}{L} \sum_l |A_l|_2^2 \tag{D.21}$$

for *some* composable sequence $A_1, \ldots, A_L$ such that $A_L \cdots A_1 = B$. As noted in (Gunasekar et al., 2018) the second inequality of eq. (D.20) is simply the arithmetic-geometric mean inequality applied to $|A_1|_2^2, \ldots |A_L|_2^2 \in \mathbb{R}_{\geq 0}$. Furthermore, we can obtain eq. (D.21) using the singular value decomposition of $B$: let $B = U S V^*$ where, letting $r := \min\{m, n\}$, $U \in U(r, m)$ and $V \in U(n, r)$

are unitary and $S = \operatorname{diag}(\lambda_1, \ldots, \lambda_r)$, where $\lambda_i \geq 0$ for all $i$. We may decompose $B$ into $L$ factors like

$$B = (US^{\frac{1}{L}}) \cdot \left( \prod_{i=1}^{L-1} S^{\frac{1}{L}} \right) \cdot (S^{\frac{1}{L}} V^*), \tag{D.22}$$

and by unitary invariance of the $C_2$ (a.k.a. Frobenius) norm,

$$|US^{\frac{1}{L}}|_2^2 + \sum_{i=1}^{L-1} |S^{\frac{1}{L}}|_2^2 + |S^{\frac{1}{L}} V^*|_2^2 = |S^{\frac{1}{L}}|_2^2 + \sum_{i=1}^{L-1} |S^{\frac{1}{L}}|_2^2 + |S^{\frac{1}{L}}|_2^2$$

$$= L |S^{\frac{1}{L}}|_2^2. \tag{D.23}$$

We now note that *by definition 4.6*,

$$\left( \|B\|_{\frac{2}{L}}^S \right)^{\frac{2}{L}} = \sum_i |\lambda_i|^{\frac{2}{L}} \text{ and on the other hand}$$

$$|S^{\frac{1}{L}}|_2^2 = \sum_i |\lambda_i|^{\frac{1}{L} \cdot 2}, \tag{D.24}$$

so that combining eqs. (D.23) and (D.24) gives eq. (D.21) for the composable sequence of eq. (D.22).

It remains to prove the first inequality of eq. (D.20) — this is a special case of the non-commutative generalized Hölder inequality lemma 4.11. In fact we will prove a slightly more general statement — to state it we need a couple more definitions:

**Definition D.25.** For a (not necessarily square) complex matrix $A \in M(m \times n, \mathbb{C})$,

$$|A| := \sqrt{A^* A} \tag{D.26}$$

where $A^*$ is the conjugate transpose of $A$.

The matrix $|A|$ is Hermitian and positive semi-definite, and often referred to as the *polar* part of $A$.[13] Given any Hermitian matrix $H$ and complex number $z$, we may form the matrix $H^z$; explicitly it can be defined as $U \operatorname{diag}(\lambda_i^z) U^*$ where $H = U \operatorname{diag}(\lambda_i) U^*$ is a diagonalization of $H$. We next define unitary invariant norms on spaces of matrices. For technical reasons to be encountered shortly, we actually introduce *families of norms* compatible with the natural inclusions $M(m \times n, \mathbb{C}) \subseteq M(m' \times n', \mathbb{C})$ for $m' \geq m, n' \geq n$.

**Definition D.27** (cf. (Bhatia, 1996, §IV))**.** A **compatible family of matrix norms** is a function

$$\|-\| : \coprod_{m,n \in \mathbb{N}} M(m \times n, \mathbb{C}) \to \mathbb{R}_{\geq 0} \tag{D.28}$$

such that

   (i) the restriction of $\|-\|$ to $M(m \times n, \mathbb{C})$ is a norm (in the sense of functional analysis) for all $m, n \in \mathbb{N}$, and

   (ii) whenever $m' \geq m, n' \geq n$ and $\iota : M(m \times n, \mathbb{C}) \subseteq M(m' \times n', \mathbb{C})$ is the "upper left block inclusion" sending

$$A \mapsto \begin{bmatrix} A & 0 \\ 0 & 0 \end{bmatrix} \tag{D.29}$$

   we have $\|\iota(A)\| = \|A\|$.

A compatible family of norms is **unitary invariant** if and only if for any $A \in M(m \times n, \mathbb{C})$ and any unitary matrices $U \in U(m)$ and $V \in U(n)$

$$\|UAV^T\| = \|A\|. \tag{D.30}$$

We will show below that for any $p > 0$ the Schatten $p$-norms form a unitary invariant compatible family.

---

[13]E.g. in the polar decomposition (Higham, 2008, §8).

**Lemma D.31.** *If $B \in M(m \times n, \mathbb{C})$ is a matrix with complex entries, $A_1, \ldots, A_L$ is a composable sequence of complex matrices such that $A_L \cdots A_1 = B$ and $\sum_i \frac{1}{p_1} = \frac{1}{r}$ where $p_1, \ldots, p_L > 0$ are positive real numbers, then for every unitary invariant compatible family of norms $\|-\|$,*

$$\||B|^r\|^{\frac{1}{r}} \leq \prod_i \||A_i|^{p_i}\|^{\frac{1}{p^i}}. \tag{D.32}$$

We note that this non-commutative generalized Hölder inequality is "non-commutative" since we work with products of matrices as opposed to inner products of vectors, and "generalized" since we consider $\ell_p$ exponents $p_i$ where $\sum \frac{1}{p_i} > 1$. To be specific, lemma 4.11 is the case of lemma D.31 where the unitary invariant compatible family of norms $\|-\|$ consists of the $C_1$ norms[14] and $p_i = 2$ for all $i$. Here we are implicitly using the relationship between Schatten norms

$$\|A\|_p^S = (\||A|^p\|_1^S)^{\frac{1}{p}} \text{ for all } A \in M(m \times n, \mathbb{C}), p > 0. \tag{D.33}$$

As mentioned in section 4, when $L = 2$ lemma D.31 is (Bhatia, 1996, Exercise IV.2.7). In the case of Schatten norms, it is essentially derived in the course of the proof of (Dai et al., 2021, Thm. 1). Below, we solve (Bhatia, 1996, Exercise IV.2.7) and show that the case $L > 2$ follows by induction.[15]

We first address a subtle difference between lemma D.31 and the setup of (Bhatia, 1996) (and indeed most work on matrix analysis). (Bhatia, 1996) considers square matrices throughout, whereas in lemma D.31 we allow all matrices to be rectangular — this is essential in our applications, since neural networks have variable width (even if we made our CNNs of section 3 "constant width" by requiring $C_1 = C_2 = \cdots = C_{L-1}$, the first and last layers would still change width in general). Thankfully, there is a simple trick that allows us to reduce to the case of square matrices. Say $A_l \in M(m_l \times n_l)$ as in definition D.16, and let

$$N := \max\{m_1, \ldots, m_L, n_1, \ldots, n_L\} \tag{D.34}$$

Since $N \geq m_l, n_l$ by definition, for each matrix $A_l$ we may define a new block diagonal matrix

$$\tilde{A}_l := \begin{bmatrix} A_l & 0 \\ 0 & 0 \end{bmatrix} \tag{D.35}$$

(that is, we push $A_l$ to the top left corner). Note that by definition D.16 and the hypotheses of lemma D.31 $m = m_L, n = n_1$ and a straightforward inspection of the mechanics of block diagonal matrix multiplication shows

$$\tilde{A}_L \cdot \tilde{A}_{L-1} \cdots \tilde{A}_1 = \begin{bmatrix} B & 0 \\ 0 & 0 \end{bmatrix} =: \tilde{B} \tag{D.36}$$

Now $\tilde{A}_1, \ldots, \tilde{A}_L$ and $\tilde{B}$ are all square, and to reduce to the square case it will suffice to argue that

$$\||\tilde{A}_i|^{p_i}\| = \||A_i|^{p_i}\|^{\frac{1}{p^i}} \text{ for all } i \text{ and } \||\tilde{B}|^r\| = \||B|^r\| \tag{D.37}$$

*Since by hypothesis $\|-\|$ is a compatible family of unitary invariant norms,* it will suffice to show that

$$|\tilde{A}_i|^{p_i} = \begin{bmatrix} |A_i|^{p_i} & 0 \\ 0 & 0 \end{bmatrix} \tag{D.38}$$

and this is follows from the identities

$$\begin{bmatrix} A & 0 \\ 0 & 0 \end{bmatrix}^* \begin{bmatrix} A & 0 \\ 0 & 0 \end{bmatrix} = \begin{bmatrix} A^*A & 0 \\ 0 & 0 \end{bmatrix} \text{ and } \begin{bmatrix} H & 0 \\ 0 & 0 \end{bmatrix}^z = \begin{bmatrix} H^z & 0 \\ 0 & 0 \end{bmatrix} \tag{D.39}$$

valid for any complex matrix $A$, Hermitian matrix $H$ and complex number $z$, the proofs of which we omit.

Before continuing with the proof of lemma D.31, we pause to verify that Schatten norms are indeed a unitary invariant compatible family. In doing so we prove a lemma that will be of further use in the sequel.

---

[14]Which is simply the sum of the singular values (sometimes called the *trace norm*).

[15]We beg forgiveness for posting a Springer GTM exercise solution on the internet.

**Definition D.40.** A **compatible family of guage functions** is a function $\Phi : \coprod_n \mathbb{R}^n \to \mathbb{R}_{\geq 0}$ such that

(i) for each $n \in \mathbb{N}$ the restriction of $\Phi$ to $\mathbb{R}^n$ is a norm (in the sense of functional analysis) and

(ii) whenever $n' \geq n$ and $\iota : \mathbb{R}^n \to \mathbb{R}^{n'}$ is the inclusion mapping $(x_1, \ldots, x_n) \mapsto (x_1, \ldots, x_n, 0, \ldots, 0)$, we have $\Phi(\iota(x)) = \Phi(x)$.

A **compatible family of guage functions** is *symmetric* if and only if for each $n \in \mathbb{N}$ the restriction of $\Phi$ to $\mathbb{R}^n$ is invariant under the action of matrices of the form $PD$ where $P$ is a permutation and $D$ is diagonal, with diagonal entries in $\{\pm 1\}$.

**Lemma D.41** (cf. (Bhatia, 1996, Thm. IV.2.1)). *There is a natural one-to-one correspondence between unitary invariant compatible families of matrix norms and symmetric compatible families of guage functions.*

*Proof.* Given a unitary invariant compatible family of matrix norms $\|-\|$, a symmetric compatible family of guage functions $\Phi$ can be defined using the maps

$$\mathbb{R}^n \xrightarrow{\text{diag}} M(n \times n, \mathbb{C}) \xrightarrow{\|-\|} \mathbb{R}_{\geq 0}; \tag{D.42}$$

compatibility of $\Phi$ comes from compatibility of $\|-\|$ and the identity "diag $\circ \iota = \iota \circ$ diag" (suitably interpreted), and symmetry of $\Phi$ follows from the identity $\text{diag}(PDx) = (PD) \text{diag}(x)(PD)^*$ for matrices $PD$ as above, the fact such matrices $PD$ are unitary and unitary invariance of $\|-\|$.

Conversely, given a symmetric compatible family of guage functions $\Phi$ one may define a unitary invariant compatible family of matrix norms $\|-\|$ as $\|A\| := \Phi(s(A))$ where $s(A)$ denotes the singular values of $A$. Compatibility comes from the fact that the singular values of a block diagonal matrix

$$\begin{bmatrix} A_1 & 0 \\ 0 & A_2 \end{bmatrix}$$

are the concatenation of $s(A_1)$ and $s(A_2)$, and unitary invariance follows from the fact that singular values unitary invariant up to permutations (and $\Phi$ is symmetric). The proof that the maps $\|-\| : M(m \times n, \mathbb{C}) \to \mathbb{R}_{\geq 0}$ are indeed *norms* is as in (Bhatia, 1996, Thm. IV.2.1).

It can be verified that these maps are mutual inverses — we omit this final step. $\square$

**Corollary D.43.** *For any $p > 0$ the Schatten $p$-norms form a unitary invariant compatible family of matrix norms.*

*Proof.* By *definition*, $\|A\|_p^S = \left(\sum_i s(A)_i^p\right)^{\frac{1}{p}}$, i.e. the $\ell_p$ norm of the singular values of $A$. By lemma D.41 it suffices to show that the $\ell_p$-norms form a symmetric compatible family — this is straightforward and omitted. $\square$

We now resume proving lemma D.31, first in the case $L = 2$ (later we will prove the general case by induction). Recall that at this point we have reduced to the case where all matrices in sight are $n \times n$ for some fixed $n \in \mathbb{N}$, so in particular we are dealing with a fixed unitary invariant norm (no further need for compatible families). By the above lemma, $\|A\| = \Phi(s(A))$ for some symmetric guage function $\Phi$, for all $A \in M(n \times n, \mathbb{C})$. By (Bhatia, 1996, Thm. IV.2.5),

$$s(A_1 A_2)^r <_w s(A_1)^r s(A_2)^r \tag{D.44}$$

where $<_w$ denotes weak submajorization. Now the "strongly isotone" property of the symmetric guage function $\Phi$ implies

$$\Phi(s(A_1 A_2)^r) \leq \Phi(s(A_1)^r s(A_2)^r) \tag{D.45}$$

We need a generalized Hölder inequality for symmetric guage functions.

**Lemma D.46** ((Bhatia, 1996, Ex. IV.1.7)). *If $p_1, p_2 > 0$ and $\frac{1}{p_1} + \frac{1}{p_2} = \frac{1}{r}$ then for every symmetric guage function $\Phi : \mathbb{R}^n \to \mathbb{R}_{\geq 0}$ and every $x, y \in \mathbb{R}^n$ we have*

$$\Phi(\text{abs}(x \cdot y)^r)^{\frac{1}{r}} \leq \Phi(\text{abs}(x)^{p_1})^{\frac{1}{p_1}} \Phi(\text{abs}(y)^{p_2})^{\frac{1}{p_2}} \tag{D.47}$$

*where* abs *denotes the coordinatewise absolute value.*

*Proof.* Apply the regular $r = 1$ Hölder inequality (Bhatia, 1996, Thm. IV.1.6) to the vectors $\tilde{x} = \text{abs}(x)^r, \tilde{y} = \text{abs}(y)^r$ (coordinatewise $r$-th powers) with the exponents $\tilde{p_1} = p_1/r, \tilde{p_2} = p_2/r$ (note that $\frac{1}{\tilde{p_1}} + \frac{1}{\tilde{p_2}} = \frac{r}{p_1} + \frac{r}{p_2} = 1$) to obtain

$$\Phi(\text{abs}(x \cdot y)^r) = \Phi(\text{abs}(\tilde{x} \cdot \tilde{y})) \le \Phi(\text{abs}(\tilde{x})^{\tilde{p_1}})^{\frac{1}{\tilde{p_1}}} \Phi(\text{abs}(\tilde{y})^{\tilde{p_2}})^{\frac{1}{\tilde{p_2}}} = \Phi(\text{abs}(x)^{p_1})^{\frac{r}{p_1}} \Phi(\text{abs}(y)^{p_2})^{\frac{r}{p_2}}$$
(D.48)

where in the last equality we have just used the definitions of $\tilde{x}, \tilde{y}, \tilde{p_1}$ and $\tilde{p_2}$. Taking $r$-th roots completes the proof. $\square$

Now applying lemma D.46 to eq. (D.45) gives

$$\Phi(s(A_1 A_2)^r)^{\frac{1}{r}} \le \Phi(s(A_1)^r s(A_2)^r)^{\frac{1}{r}} \le \Phi(s(A_1)^{p_1})^{\frac{1}{p_1}} \Phi(s(A_2)^{p_2})^{\frac{1}{p_2}}$$
(D.49)

(the first inequality is just taking $r$-th roots of eq. (D.45), the second is applying lemma D.46). Using the identity $s(|A|^r) = s(A)^r$, we finally obtain

$$\||A_1 A_2|^r\|^{\frac{1}{r}} \le \||A_1|^{p_1}\|^{\frac{1}{p_1}} \||A_2|^{p_2}\|^{\frac{1}{p_2}}$$
(D.50)

which is lemma D.31 when $L = 2$.

Now suppose $L > 2$ and assume by inductive hypothesis that lemma D.31 holds for all smaller values of $L$. Define

$$\frac{1}{q} = \sum_{i=1}^{L-1} \frac{1}{p_i}$$
(D.51)

(note that $\frac{1}{p_L} + \frac{1}{q} = \frac{1}{r}$). By the $L = 2$ case of lemma D.31

$$\||A_L \cdot (A_{L-1} \cdots A_1)|^r\|^{\frac{1}{r}} \le \||A_L|^{p_L}\|^{\frac{1}{p_L}} \||A_{L-1} \cdots A_1|^q\|^{\frac{1}{q}}$$
(D.52)

and by inductive hypothesis

$$\||A_{L-1} \cdots A_1|^q\|^{\frac{1}{q}} \le \prod i = 1^{L-1} \||A_i|^{p_i}\|^{\frac{1}{p_i}}.$$
(D.53)

