# OpenReview forum: "Data dependent frequency sensitivity of convolutional neural networks"
_ICLR.cc/2023/Conference — Submitted to ICLR 2023_

### Official Review · Reviewer_pf5x · 2022-10-24

**Confidence:** 4
**Correctness:** 3
**Technical Novelty And Significance:** 2
**Empirical Novelty And Significance:** 2
**Recommendation:** 5

**Clarity, Quality, Novelty And Reproducibility:**

- The use of the notation $\cdot$ (`dot`) is confusing and non-standard. The Fourier transform of a `multi-channel` convolution is not pointwise multiplication, but it is a matrix algebra with entries of the matrix being a linear operator (==convolution in the paper). Though the authors correctly defined such mapping, the `dot` notation shouldn't be used here as it is non-standard.

- Definition 4.6. Why introduce a new notation $C_p$ norm? As you mentioned, this is the Schatten p-norm and which is standard terminology.

**Strength And Weaknesses:**

# Strength

- The problem studied in this paper is quite interesting.
- The paper is relatively easy to follow.

# Weaknesses

- The setting is quite simple. The authors study linear convolutional networks, which is not that different from linear models.

- Even for linear convolutional networks, the analysis is only restricted to a couple of simple cases (Theorem 4.9). E.g., channel size all equal to 1 or the depth L\leq 2. The general, more practical setting (e.g., multiple channels), which required new ideas beyond basic Fourier transforms, is not covered in the paper.

- The reliance on Fourier transform also requires the kernel size ( 3 by 3 in practice) to be the same as the image size (e.g., 32 by 32 for Cifar 10). This restriction reduced the interest and relevance of the approach. Finding minimizers in the frequency domain



**Summary Of The Paper:**

In this paper, the authors study the sensitivity of convolutional networks (CNNs) to perturbations in frequency components of input images. The authors consider linear convolutional networks. By applying the Fourier transform, in some simple settings, the problems can be reduced to,  essentially, a linear regression (or lasso). Then, tools from linear models (e.g. eigen-decomposition) can be applied to compute the closed-form predictor, which can be used to analyze how sensitive the linear predictor is to perturbation in different Fourier modes.

**Summary Of The Review:**

update

I thank the authors for improving the paper. In particular, the current form of theorem 4.9, if correct, is much stronger than the previous version. As such, I increase my score accordingly. However, I am not comfortable accepting the paper at the moment for the reasons below. 1. the paper is still in the linear convolutional setting, aka, Tibshirani, Gunasekar, etc., though the improvement is not as obvious as the previous version. 2. There are some significant changes in the main text of the paper, as well as the main result Theorem 4.9. Because of so, I would prefer the paper to go through a new review process with the proof of Lemma 4.11 carefully checked.


------------------------------


Overall, the paper has several interesting observations. However, the contribution is incremental and does not go beyond existing work much, e.g., Tibshirani, Gunasekar, etc. The setting only covers some simple cases of linear convolutional networks. The obtained insights (e.g., sparsity, the predictor is more sensitive to low-frequency mode) seem not that new and can be derived from (deep) linear models.

---

> ### Author Response · Authors · 2022-11-17
> **Linear vs. non-linear CNNs, multiple channels, novelty**
>
> Thank you very much for your careful consideration of our paper and thoughtful comments.
>
> The purpose of the experiments section is to test whether conclusions drawn from
> our theorem about simplified CNNs remain qualitatively true for some real CNNs
> (with nonlinearities and small kernels, pooling etc.). More broadly speaking, we
> view the fact that our toy CNNs are not that different from linear models as a
> feature, not a bug: it allows for applying data-dependent shrinkage properties
> of sparse linear models. Certainly there are many aspects of modern CNNs that
> will not admit explanations in terms of linear CNNs, but should we not be happy
> when linear CNNs provide insight?
>
> Restrictions in the main theorem have been lifted -- please see our global
> comments.
>
> Regarding small kernels, please see our global comments.
>
> We are sympathetic to your complaint about the $\cdot$ notation -- would you
> prefer plain juxtaposition? We considered this but it also seems undesirable as
> the objects in question are not matrices. We entirely agree that the widely used terminology
> of Schatten $p$-norms is preferable.
>
> Regarding your comment
> >"The obtained insights (e.g., sparsity, the predictor is more sensitive to
> >low-frequency mode) seem not that new and can be derived from (deep) linear
> >models":
>
> Do you have a reference in mind where this has been done before, in such a way
> that one could easily deduce our results for spatial frequency statistics of
> real image data? Our expanded related work section in Appendix A cites some work
> on model sensitivity to low frequency modes, however they are all technically
> quite distinct from what we do here and hence do not apply in any direct way
> to give the results of our paper.

---

> ### Author Response · Authors · 2022-11-30
> **Thank you for considering our revision!**
>
> Thanks very much for taking the time to review our revision, and for your
> encouragement to further pursue this research. We realize that due to the size
> of our rebuttal revision and the necessity of further discussion, it makes more
> sense to resubmit elsewhere.  In particular we would also welcome scrutiny of the proof of Lem. 4.11.

---

### Official Review · Reviewer_bWYa · 2022-10-24

**Confidence:** 4
**Correctness:** 2
**Technical Novelty And Significance:** 3
**Empirical Novelty And Significance:** 3
**Recommendation:** 5

**Clarity, Quality, Novelty And Reproducibility:**

The paper reads well and is clear for the most part. I think the novelty is limited since the main theorem does not go much beyond the straight forward extensions of known theorems, and the empirical results do not really support the theoretical extensions or the main claim of the paper (as detailed above). The results appear to be sufficiently reproducible.

**Typos**

The formatting is not for ICLR 2023.
1/p_i in Conjecture 4.13.
Figure 6 missing caption.

**Strength And Weaknesses:**

**Strengths**

The paper posits an interesting theorem, and motivates it sufficiently well with some prior evidence. The writing is mostly clear, although the format is wrong. While I don’t think the main claim of the paper is correct, the much more restricted claim that is actually proved is interesting and can help explain some observations in practice.

**Weaknesses**

I have the following concerns/questions about the paper:

1) The main result in theorem 4.9 concerns a very specific case of linear CNNs, and as the paper itself points out is a quite straightforward extension of Tibshirani 2021. Now this in itself is not an issue, what becomes an issue is that the paper does not go much beyond this extension, for example considering the effect of size-limited filters and also the more complicated case of simple ReLU non-linearities, which would make the theory much more substantial.

2) The theoretical results really show the sensitivity caused by the norm penalties, rather than the CNN itself, since removing the penalties (lambda=0) will result in a “non-sensitive” CNN. In other words, the simplification of CNN structure has stripped it off its many inductive biases, and as a result exposing the inductive bias due to norm penalties, which are well-known already (although not explicitly discussed in the spectrum as far as I know). So it is misleading to claim “CNNs inherit frequency sensitivity from image statistics” where the theory only shows this happening due to norm penalties when CNN is extremely simplified.

3) As the paper itself points out in Section 6, the empirical experiments only on a low-frequency prominent dataset (CIFAR) do not provide any evidence for the main claim of the paper: “CNNs inherit frequency sensitivity from image statistics”. Repeating these experiments on datasets where different frequency bands are prominent (which are trivial to construct) are a strict requirement to support the main claim. Also, the empirical results should always report error bars to show statistical significance of the results (no figure currently shows error bars).


**Summary Of The Paper:**

This work proposes a theoretical explanation for the observation of imbalanced frequency sensitivity in CNNs. The paper studies linear CNNs under weight penalties, and derives justification for the dependence of such CNNs' frequency sensitivity on the distribution of power in the data spectrum. Two empirical experiments involving actual non-linear CNNs trained on CIFAR are provided to support some predictions of the proposed theory, namely that adding more layers and increasing weight decay coefficient will both emphasize frequency sensitivity of CNNs to lower frequency components.

**Summary Of The Review:**

The paper provides a theory explaining how linear CNNs under certain assumptions and weight penalties tend to have an inductive bias reflecting the spectral composition of the input data. However, the very restricted theorem, and lack of extensive empirical experiments supporting the theorem and the main claim, limits the paper’s significance and validity.

---

> ### Author Response · Authors · 2022-11-17
> **What's special about CNNs, norm penalties, experiments**
>
> Thank you very much for your thorough read of our paper and insightful
> commentary and suggestions.
>
> Regarding analysis of CNNs with nonlinearities and small kernels, please see our
> global comments.
>
> After applying the DFT CNNs essentially diagonalize in frequency space, and this
> is what allows us to make a connection with frequency statistics of natural
> images (in particular we use the fact that the covariance matrix of natural
> image data is not far from diagonal in DFT frequency space).
>
> In this way we do make essential use of *architecture* properties of CNNs. Note
> also that the representational cost of an MLP in the same situation would be
> quite different: for simplicity, in the case of a binary classification task
> (output dimension $K=1$), on image data with a single channel ($C=1$) a linear
> MLP with weights $W_1, \dots, W_L$ and linear predictor $B = W_L \cdots W_1$
> would have an effective regularization penalty $\lambda L (\lVert B
> \rVert_{2/L}^S)^{2/L}$. With the simplification $C=K=1$, $B$ is essentially a
> vector of length $H W$ and so it has 1 singular value $\lVert B \rVert_2$
> (regular Euclidean norm of the corresponding vector), hence $\lambda L (\lVert B
> \rVert_{2/L}^S)^{2/L} = \lambda L \lVert B \rVert_2^{2/L}$. In contrast, with
> the same simplifying assumptions ($C=K=1$) the effective regularization penalty
> for an L-layer CNN with linear predictor $B$ is $\lambda L \sum_{i,j}
> B_{ij}^{2/L}$. Note for example that the MLP regularization penalty is invariant
> to preprocessing with arbitrary orthogonal rotations of input data (including
> along the spatial indices), whereas in general the penalty for the CNN is only
> invariant under preprocessing with permutations and sign flips of the frequency
> components.  We hope the above provides evidence that our analysis says
> something about CNNs, specifically.
>
> We agree that regularization is an essential part of our theoretical and
> empirical results (although it can be argued that given the ubiquity of weight
> norm regularization either via decay or early stopping, the conclusions of our
> analysis likely apply quite broadly). On a related note, we concur that the
> claim made in the title was more broad than what was proved/demonstrated
> empirically and have updated the title to be more precise.
>
> Regarding experiments on image datasets with different frequency content, we
> completely agree; see our global comments for a summary of new experiments.
>
> The purpose of our experiments is to show that the implications of our main
> theorem remain qualitatively valid for (at least some) CNNs with nonlinearities
> and small kernels. Moreover, note that even with extremely small weight decay the
> frequency sensitivity curves reflect the frequency content of the training data
> and hence illustrate that (again for some real-world CNNs) model frequency
> sensitivity is data dependent even with negligible weight decay.
>
> Regarding significance, please see our global comments.

---

> > ### Comment · Reviewer_bWYa · 2022-11-28
> > **Response to revisions**
> >
> > Thank you for your revisions and explanations. The new theorem and the additional experiments indeed make the paper's contribution more substantial, as such I'm increasing my score to 5. However, I think the experiments on high-pass filtered images are not consistent with the predictions of the theorem. In particular, in Figures 1, 5, 6, 7, although the sensitivity increases in the high pass region in some cases (except in 5), **almost always the CNNs are very sensitive to low frequencies**, which is in contrast to the theorem given that the low frequencies are mostly filtered out. The current discussion of the results are not nuanced enough and does not discuss these interesting results in enough detail. Given the substantial changes made to the paper which requires in-depth discussion and perhaps even refinement of the current theorem to account for the new observations, I cannot suggest acceptance at present. Nonetheless, I think the contributions are valuable, and encourage the authors to discuss the newly added experiments in more depth and resubmit to the next venue.

---

> > > ### Author Response · Authors · 2022-11-30
> > > **Sensitivity to the lowest frequencies even after high pass filtering does indeed require further analysis/discussion**
> > >
> > > Thanks very much for taking the time to review our revision, and for your
> > > encouragement to pursue this research further. We realize that due to the size
> > > of our rebuttal revision and the necessity of further discussion, it makes more
> > > sense to resubmit elsewhere.
> > >
> > > The fact that essentially accross the board sensitivity in the lowest
> > > frequencies remains even after training on high-pass filtered images is indeed
> > > unexpected from the view of our theoretical analysis, and warrants much more
> > > thorough and nuanced discussion. For now, a couple comments/speculations:
> > > - We do want to avoid clinging to bias towards confirmation of our main theorem,
> > >   but in all of the mentioned plots the y-axes for frequency sensitivity of
> > >   models trained on natural vs high pass filtered images are distinct, and on a
> > >   log scale. In figures 1 and 6 we see significant drop in sensitivity to the
> > >   lowest frequencies, so that while models trained on high-pass filtered images are still sensitive in the lowest frequencies, they are significantly less sensitive there than models trained on natural images. In figure 5 the drop does not appear to be statistically
> > >   significant, and in figure 7 it actually appears that the models trained on
> > >   high-pass filtered images are *more* sensitive to the lowest frequencies -- in general our
> > >   experimental results for ResNets suggest they do not conform to the
> > >   predictions derived from Thm. 4.9 which is not surprising as our idealized
> > >   CNNs don't have residual connections.
> > > - It is indeed true that if the CNNs in our experiments had data dependent frequency
> > >   sensitivities qualitatively described by  Prop. 4.18 and eq. 4.19, those
> > >   trained on high-pass filtered images would have no sensitivity to the lowest frequencies.
> > >   There are a number of possible reasons why we don't see this:
> > >   - The simplest is that our idealized CNNs are very different from those
> > >     occuring in the experiments. We plan to include some discussion of the known
> > >     frequency space effects of small convolutional kernels, pooling etc. in
> > >     further revisions.
> > >   - Another is that our theoretical analysis doesn't take into account gradient
> > >     descent dynamics. While our references on implicit bias suggest that at
> > >     classifiers trained with SGD and cross entropy loss a statement analogous to
> > >     Thm. 4.9 is true (we are thinking of eq. A.1 here), such a statement would
> > >     only apply at *convergence* of SGD. One very simple experiment we would like to
> > >     add is computations of frequency sensitivity curves of *randomly
> > >     initialized* CNNs in all relevant figures.

---

### Official Review · Reviewer_yP96 · 2022-10-24

**Confidence:** 5
**Correctness:** 2
**Technical Novelty And Significance:** 2
**Empirical Novelty And Significance:** Not applicable
**Recommendation:** 3

**Clarity, Quality, Novelty And Reproducibility:**

Novelty and significance is below the standard for ICLR.


**Strength And Weaknesses:**

Strength: lots of derivations.

Weakness: lack of experiments and competing results.

One fundamental issue: it has been long known that CNNs train better with whitened images -- which is equivalent to equalizing the images in the frequency domain beforehand. There is no room for sensitivity once proper preprocessing is done. The analysis of this paper seems irrelevant and may be in a wrong direction.


**Summary Of The Paper:**

It is unclear what concrete contributions of this paper are. The paper does have quite a lot of equations, but it is hard to say that the derivations are upon a good problem. Experiments are severely lacking.

**Summary Of The Review:**

I vote for rejection because the significance of the analysis is low.

---

> ### Author Response · Authors · 2022-11-17
> **Concrete contributions, purpose of derivations, impact of image "whitening"**
>
> Thank you for this feedback.
>
> Clarification on our concrete contributions has been added to the last paragraph
> of the intro (please see also our global comments).
>
> The purpose of our derivations is to identify bias on a CNN imposed by a
> learning objective with $\ell_2$ regularization. This purpose is in part similar to the
> purpose of many articles on implicit bias and representational cost (e.g. those
> in our references), many of which have appeared at major conferences. At the
> same time, we go further than most such papers by analyzing *data-dependent*
> bias. We argue that analyzing the impact of training data statistics on the type
> of representation learned by a CNN is a worthwhile pursuit.
>
> We agree that the experiments were lacking, and have expanded them along
> multiple axes. Please see our global comments for a summary of new material.
>
> Standard image "whitening" normalizes the statistics RGB values. It will *not*
> result in standardized variances along DFT frequency components; in fact, it
> only affects 0th order DFT statistics. Moreover, all experiments in this paper
> were conducted using standard "whitening" preprocessing, so if such
> preprocessing isolated CNNs from the frequency statistics of image datasets, our
> results would have looked quite different. Appendix B now contains further
> discussion of this point.
>
> In light of the above, we ask that you please reconsider the relevance and
> directional correctness of our paper.

---

> > ### Comment · Reviewer_yP96 · 2022-11-17
> > **Pixel-wise whitening was a standard preprocessing step.**
> >
> > The author should definitely check the literature, e.g. PCA whitening as preprocessing for pixels in the CIFAR-10/100 times.

---

> > > ### Author Response · Authors · 2022-11-17
> > > **Apologies for misunderstanding your earlier comment about ZCA, and thank you for a more precise reference.**
> > >
> > > Would [1-4] be the sort of papers you are referring to? If so, yes they are definitely worth us looking at.
> > >
> > > A couple further remarks:
> > > - ZCA whitening is still not literally equivalent to equalizing the images in the frequency domain beforehand, although of course it is very similar based on empirical calculations of natural image covariance matrices (e.g. fig. 4 in the current revision)
> > > - ZCA whitening would be another type of preprocessing worth including in our experiments section. We will consider doing so in a future revision (though it will not be possible to do so before tomorrow evening).
> > > - Supposing it is the case that radial frequency sensitivity curves of CNNs trained on ZCA whitened images are relatively flat (this is what our theory would predict, but without doing experiments we can only hypothesize), it would not contradict the main claim of this paper. It would just be a case in which CNNs inherit flat radial frequency curves from input images.
> > > - While based on our reading of the references below you are correct that CNNs train better with whitened images, having looked at many articles and codebases we still see few if any e.g. Imagenet training algorithms implementing full per pixel ZCA whitening. Hence it is still unclear how these questions regarding the effect of ZCA preprocessing render our paper irrelevant.
> > >
> > > [1] A. Krizhevsky, “Learning Multiple Layers of Features from Tiny Images,” undefined, 2009, Accessed: May 12, 2022. [Online]. Available: https://www.semanticscholar.org/paper/Learning-Multiple-Layers-of-Features-from-Tiny-Krizhevsky/5d90f06bb70a0a3dced62413346235c02b1aa086
> > >
> > > [2] A. Coates, A. Ng, and H. Lee, “An Analysis of Single-Layer Networks in Unsupervised Feature Learning,” in Proceedings of the Fourteenth International Conference on Artificial Intelligence and Statistics, Jun. 2011, pp. 215–223. Accessed: Nov. 17, 2022. [Online]. Available: https://proceedings.mlr.press/v15/coates11a.html
> > >
> > > [3] L. Huang, D. Yang, B. Lang, and J. Deng, “Decorrelated Batch Normalization.” arXiv, Apr. 23, 2018. Accessed: Nov. 17, 2022. [Online]. Available: http://arxiv.org/abs/1804.08450
> > >
> > > [4] J. Lee et al., “Finite Versus Infinite Neural Networks: an Empirical Study.” arXiv, Sep. 08, 2020. doi: 10.48550/arXiv.2007.15801.

---

> > > > ### Comment · Reviewer_yP96 · 2022-11-17
> > > > **Frequency whitening is easy.**
> > > >
> > > > These are some related references.
> > > > It is easy to whiten the image in the frequency domain before the learning and the learning is just fine.
> > > > In this case would the authors claim any useful knowledge in the paper?

---

> > > > > ### Author Response · Authors · 2022-11-18
> > > > > **Degree of difficulty of frequency whitening is orthogonal to our paper.**
> > > > >
> > > > > Great, thanks for confirming the relevance of those references.
> > > > >
> > > > > Yes, we would claim useful knowledge in that case. For example, one could easily design a dataset $\mathcal{D} = \{(x_n, y_n\}$ in which the images $x_n$ have been whitened in the frequency domain, but due to correlations between the inputs and the labels $y_n$, in the absence of any regularization a CNN trained on $\mathcal{D}$ would have sensitivity to certain frequencies. For example, make the learning task "is this image a solid color", i.e. does it align with the 0th order DFT basis vector. Our theory implies (for the CNNs defined in Sec. 3) that increasing regularization (or alternatively, in the presence of a little weight decay increasing depth) would flatten the radial frequency curve of the CNN. Is the impact of depth/decay in this situation *a priori* obvious? Is understanding factors that reduce a CNNs response to label correlation useless? Of course, since we haven't run this specific experiment we do not have empirical results to supply.
> > > > >
> > > > > Would you mind clarifying what you mean by "the learning is just fine"? We do not claim in this paper that for example the low frequency bias of natural image datasets and CNNs trained on them is a problem that needs to be fixed. We do claim that there have been a number of empirical papers raising questions about the frequency sensitivities of various CNNs trained on various datasets, and that we have a novel theoretical explanation for some of their findings. We approached this paper from the perspective that knowledge explaining observed phenomena is useful.
> > > > >
> > > > > And at the risk of repeating ourselves, given the ubiquity of CNNs trained on images that have not been per-pixel ZCA whitened, the usefulness of our paper does not hinge on what it says in that particular case.
> > > > >
> > > > > **One more thought**: if the variation in CNN gradient norms across frequency magnitudes is viewed as a issue, and if we do believe per-pixel ZCA preprocessing would largely eliminate this variation, then one could view our findings as yet another argument for ZCA preprocessing.

---

### Author Response · Authors · 2022-11-17
**Revision highlights, argument for significance of this paper**

We would like to thank all the reviewers for their questions and comments, which
we feel have substantially improved the quality of this paper.

There are numerous updates in the new revision. Some are discussed in individual
replies, but here are those relevant to two or more reviews:

- **Title updated** to more precisely reflect our results.
- Last paragraph of introduction includes **clarification of contribution of this
  paper.** In particular, we acknowledge (at multiple points in the paper) that
  our theoretical results build on prior work, and our analysis of shrinkage
  properties of linear models is obviously not new -- however, *we are not aware
  of any existing work that combines implicit bias/representational cost,
  regularized linear models and empirical observations about real data
  distributions to provide insight on behavior of deep learning models.* While
  in hindsight this may seem like a straightforward connection to make, it is
  surprising that few if any implicit bias/representational cost papers include
  applications to real data, and that few if any papers on frequency sensitivity
  of CNNs include any application of such theoretical results, which as we show
  are quite relevant. As such, we argue that this perhaps straightforward
  connection is both novel and entirely worth pointing out.
- Conjecture on non-commutative generalized Hölder has been upgraded to Lemma 4.11, and
  as a consequence the **restrictions on Theorem 4.9 have been removed**. This means
  the proofs in Appendix D have been expanded (and considerably cleaned up).
  While it is fair to characterize this theorem as an extension of
  pre-existing results, the extension is non-trivial, and we argue that the
  connection between representation cost of deep linear models and
  non-commutative generalized Hölder inequalities is an observation of broader
  interest that has not appeared elsewhere to the best of our knowledge.
- Experiments have been expanded to include **more datasets** (ImageNette and
  synthetic data generated by a Wavelet Marginal Model), **more variable
  frequency statistics** (obtained by high pass filtering CIFAR and ImageNette),
  and **more CNNs** (VGGs, AlexNets). In addition, we include repeated runs
  throughout to obtain error bars. As a byproduct the appendix now includes more
  experimental details.
- Lastly, multiple reviewers noted the simplicity of our theoretical framework.
  We agree that our theoretical setting is quite simple. However, one can argue
  that this is a feature not a bug: indeed, we show that it can be used to
  derive  predictions that carry over to (some) CNNs with small kernels and
  ReLUs. As pointed out above, our goal is to explain a phenomena (data
  dependent frequency sensitivity in CNNs). It seems reasonable to do so with
  the simplest theory we can get away with. Moreover, there are legitimate
  technical reasons why we, along with almost all of our references on implicit
  bias/representation cost (many of which appeared in top conferences/journals)
  use linear CNNs with circular convolutions: the technical difficulties pile up
  when one leaves that setting.
    - As the references show (Jagadeesan among others), dealing with
      small kernels is quite challenging. It would indeed be of interest to
      analyze optimization problems in that setting, but that is not the primary
      goal of our paper.
    - Moreover, while one might hope for example that some of the analysis of
      simple ReLU networks in Sec. 4 of our Tibshirani reference could possibly
      apply to a 2 layer CNN with ReLU nonlinearities, we would need to apply that
      reference on the other side of the FFT, so where ReLUs have to be replaced
      with projection onto the nearest positive semi-definite function (this
      last part is essentially a theorem of Bochner). We had no luck with any
      such analysis.

**Note**: error bars for ConvActually and VGGs trained on ImageNette with larger high pass filter cutoff experiments will be added to the revision shortly.

**Update**: Error bars for the aforementioned ConvActually and VGG experiments now have error bars. To make this possible we did have to include fewer depths/decays in the sweeps, please see the new figures and Appendix B for those details. The plot displaying results for contrastively trained AlexNets on WMM synthetic images still lacks error bars. It would, of course, be possible to go back and train more models for the latter experiment from different initializations for a future version of this paper (we simply ran out of time).

---

### Decision · Program_Chairs · 2023-01-20

**Decision:**

Reject

**Justification For Why Not Higher Score:**

The paper was an obvious reject at submission time, due to lack of novelty and significance.

**Justification For Why Not Lower Score:**

N/A

**Metareview: Summary, Strengths And Weaknesses:**

In this work, the authors develop a theoretical analysis of the sensitivity of convolutional neural networks to perturbations in frequency.  The reviewers found the motivation and problem setting compelling.  However, a common criticism was that the theory was applied in an overly simplified setting, i.e. linear CNNs, and as a result both lacked novelty over existing work (Tibshirani, etc.) and didn't satisfy the paper's claims.  In discussion, the authors presented an expanded analysis that seemed to please two of the reviewers, who both agreed that it could be convincing and raised their scores.  However, the reviewers still did not recommend an accept, because they felt the changes were too substantial to review and verify within the scope of the review period.  As a result the review scores are 3, 5, 5.  (Note that the review with the 3 score is terse and it's not clear that their review was easy to address or led to a productive discussion.).  As it stands, it seems clear that the paper was not ready for acceptance upon submission, but appears to have improved significantly as a result of the author's changes during the discussion period.  However, the change is too big for the reviewers to validate.  Therefore, the recommendation is to reject the paper but encourage the authors to continue to develop the theory and presentation for a future submission.